# OFMU: Optimization-Driven Framework for Machine Unlearning

**Sadia Asif,  Mohammad Mohammadi Amiri**
Rensselaer Polytechnic Institute
`{asifs, mamiri}@rpi.edu`

## Abstract

Large language models deployed in sensitive applications increasingly require the ability to *unlearn* specific knowledge, such as user requests, copyrighted materials, or outdated information, without retraining from scratch to ensure regulatory compliance, user privacy, and safety. This task, known as machine unlearning, aims to remove the influence of targeted data (*forgetting*) while maintaining performance on the remaining data (*retention*). A common approach is to formulate this as a multi-objective problem and reduce it to a single-objective problem via scalarization, where forgetting and retention losses are combined using a weighted sum. However, this often results in unstable training dynamics and degraded model utility due to conflicting gradient directions. To address these challenges, we propose **OFMU**, a penalty-based bi-level optimization framework that explicitly prioritizes forgetting while preserving retention through a hierarchical structure. Our method enforces forgetting via an inner maximization step that incorporates a similarity-aware penalty to decorrelate the gradients of the forget and retention objectives, and restores utility through an outer minimization step. To ensure scalability, we develop a two-loop algorithm with provable convergence guarantees under both convex and non-convex regimes. We further provide a rigorous theoretical analysis of convergence rates and show that our approach achieves better trade-offs between forgetting efficacy and model utility compared to prior methods. Extensive experiments across vision and language benchmarks demonstrate that OFMU consistently outperforms existing unlearning methods in both forgetting efficacy and retained utility.

## 1    Introduction

Large language models (LLMs) have become foundational to applications ranging from search engines and coding assistants to healthcare, education, and scientific discovery. Their remarkable performance arises from training on massive and diverse corpora, which inevitably contain sensitive, copyrighted, or harmful information. This raises serious concerns about privacy, regulatory compliance, safety, and ethics. In particular, regulations such as the General Data Protection Regulation (GDPR) (European Union, 2016; California Legislative Counsel, 2018; Office of the Privacy Commissioner of Canada, 2018) grant individuals the "right to be forgotten" (Dang, 2021) requiring deployed models to eliminate the influence of specific data upon request. Beyond regulatory mandates, unlearning is also necessary to prevent models from generating toxic content, leaking private information, or providing instructions for misuse (Huang et al., 2022; Carlini et al., 2023; Staab et al., 2024). These considerations have led to growing interest in *machine unlearning*, the ability to selectively erase the impact of particular data from a trained model while maintaining its utility.

**Limitations of Existing Approaches.** Existing unlearning methods for LLMs can be broadly categorized into three families (see Appendix 7.6 for a broader discussion): input-based, data-based, and model-based approaches. Input-based methods modify the prompts or instructions given to the model so that it refuses to generate content related to the forget set (Pawelczyk et al., 2023; Liu et al., 2024b). These methods are lightweight but typically brittle, as adversarial prompts can often bypass the refusal policy. Data-based methods fine-tune the model on curated examples that encourage desirable outputs when queried with forget-related prompts (Choi et al., 2024). While effective in narrow settings, such methods risk semantic distortion, require careful construction of

auxiliary data, and may not generalize beyond specific domains. Model-based approaches directly alter model parameters through techniques such as fine-tuning, gradient ascent, or projection (Bu et al., 2024; Fan et al., 2025; Dong et al., 2024; Zhang et al., 2024b). These approaches are generally more effective at suppressing unwanted knowledge, but they introduce a deeper optimization challenge: balancing the trade-off between forgetting the targeted information and preserving utility on the retain set.

Most model-based methods formulate this balance as a scalarized optimization problem, where the forget loss and retain loss are combined with fixed weights into a single objective. This design leads to several shortcomings. First, static weighting fails to reflect the dynamic nature of unlearning: early optimization steps should prioritize forgetting, while later updates should shift emphasis toward restoring utility. However, fixed weights cannot adapt accordingly. Second, scalarization is inherently unstable. When the forget objective dominates, the model can collapse, leading to severe performance degradation on the retain set. When the retain objective dominates, forgetting remains incomplete and sensitive data can persist. Third, most existing algorithms perform poorly on *hard-to-unlearn samples* (see Appendix 7.5.6), where forget and retain gradients are strongly entangled. In such cases, aggressive updates on the forget set cause disproportionate collateral damage to the retain set, as evidenced by the strong coupling between sample difficulty and utility loss. Figure 1 further shows that while many methods perform adequately on *easy-to-unlearn* samples, their performance drops sharply as difficulty increases. This trend underscores their inability to maintain stable performance under challenging conditions, ultimately failing to meet the true objective of unlearning. Finally, existing approaches largely lack principled theoretical grounding, instead relying on heuristic weighting schemes that scale poorly in the high-dimensional, non-convex optimization landscapes of modern LLMs.

These challenges call for a more principled and structured approach to optimization-based unlearning, one that recognizes the asymmetric priorities of forgetting and retention. Crucially, *forgetting must take precedence*. If a model fails to unlearn harmful content, its retained capabilities are irrelevant from a safety or compliance perspective. In contrast, once forgetting is successful, utility can be gradually restored as long as the erased information does not re-emerge. This observation motivates a hierarchical optimization view, where forgetting is posed as a primary inner objective, and retention is addressed as a secondary outer goal.

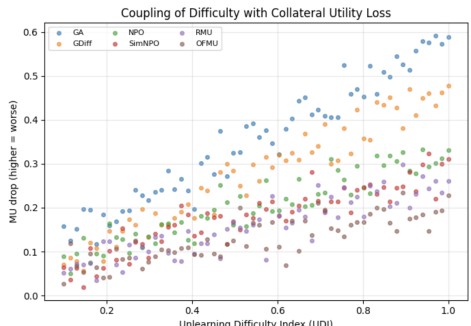

Figure 1: Coupling of unlearning difficulty with collateral utility loss. Harder samples induce disproportionately large utility degradation for existing methods (GA (Thudi et al., 2022), GradDiff (Maini et al., 2024)), whereas OFMU mitigates this coupling through its similarity-aware hierarchical updates. Full detail is provided in Appendix 7.5.6.

**Our Approach: OFMU.** We introduce **OFMU** (**O**ptimization-Driven **F**ramework for **M**achine **U**nlearning), a penalty-based bi-level optimization framework that formalizes this hierarchy. OFMU addresses the shortcomings of existing methods through three key innovations: (i) a principled penalty-based reformulation that enforces stationarity of the inner forgetting objective, enabling efficient two-loop optimization without requiring full convergence of the inner problem; (ii) a similarity-aware penalty that explicitly decorrelates gradients between forget and retain objectives, mitigating destructive interference during updates; and (iii) a rigorous convergence analysis of penalty-based unlearning under both convex and non-convex regimes.

**Contributions.** Our main contributions are summarized as follows:
- We propose OFMU, a novel optimization-driven bi-level framework that explicitly prioritizes forgetting over utility preservation, capturing the conceptual hierarchy inherent in unlearning.
- We develop a scalable two-loop algorithm with provable convergence guarantees, avoiding the computational bottlenecks of traditional bi-level optimization methods.
- We design a similarity-aware penalty that dynamically decorrelates forget and retain gradients, ensuring that forgetting does not inadvertently degrade retained knowledge.

- We provide a comprehensive theoretical analysis of convergence rates for penalty-based unlearning in both convex and non-convex settings.
- We empirically validate OFMU across benchmark unlearning tasks in language and vision models, establishing a new state-of-the-art in the forgetting-utility trade-off.

## 2   BI-LEVEL OPTIMIZATION IN MACHINE LEARNING

Bi-level optimization has emerged as a powerful paradigm for problems where two interdependent objectives must be optimized in a hierarchical manner. Formally, it consists of an *outer* optimization task whose feasible solutions are implicitly constrained by the optimal solutions of an *inner* optimization problem (Colson et al., 2007; Dempe & Zemkoho, 2020). This nested structure provides a natural way to capture tasks where one objective has strict priority over another, such as hyperparameter tuning (Franceschi et al., 2018), meta-learning (Finn et al., 2017; Nichol et al., 2018), and adversarial robustness (Madry et al., 2018).

Recent advances have demonstrated the utility of bi-level optimization in large-scale machine learning, particularly in domains where competing goals must be balanced without collapsing into trivial solutions. For instance, in meta-learning, the inner problem adapts to specific tasks, while the outer problem promotes generalization across tasks (Hospedales et al., 2021). Similarly, in adversarial training, the inner maximization crafts adversarial perturbations, and the outer minimization strengthens the model against them (Zhang et al., 2019). These successes highlight the versatility of the framework in structuring inherently asymmetric objectives and avoid trivial or unstable solutions.

We adopt this perspective for machine unlearning. In this setting, forgetting must be enforced as a non-negotiable objective, ensuring that the influence of target data is fully removed, while utility preservation is treated as a secondary goal to be optimized conditionally. This stands in contrast to scalarized approaches that conflate the two objectives via fixed weights, often leading to brittle trade-offs. Bi-level optimization, by explicitly separating the two, allows us to respect the asymmetry of their importance and design algorithms that reflect this priority structure. This foundational insight motivates our proposed framework, OFMU, which we formally present in Section 3.

## 3   METHODOLOGY

We now present the OFMU framework, a penalty-based bi-level optimization method that explicitly separates the forgetting and utility preservation objectives. We begin by introducing notation and formalizing the problem, then describe the bi-level formulation, its penalty-based reformulation, and our scalable two-loop algorithm.

### 3.1   PRELIMINARIES AND NOTATION

We consider a supervised learning setup with a dataset $\mathcal{D} = \mathcal{D}_r \cup \mathcal{D}_f$, where $\mathcal{D}_r$ is the *retain set* (examples to be preserved) and $\mathcal{D}_f$ is the *forget set* (examples to be unlearned). The model is denoted by $f_\theta$, parameterized by $\theta \in \mathbb{R}^d$.

**Empirical Losses.**   We define the empirical losses over each subset as:

$$\mathcal{L}_r(\theta) = \frac{1}{|\mathcal{D}_r|} \sum_{(x,y)\in\mathcal{D}_r} \ell(f_\theta(x), y), \quad \mathcal{L}_f(\theta) = \frac{1}{|\mathcal{D}_f|} \sum_{(x,y)\in\mathcal{D}_f} \ell(f_\theta(x), y), \tag{1}$$

where $\ell(\cdot, \cdot)$ is a standard loss function (e.g., cross-entropy).

**Gradients and Similarity.**   We denote the gradients of the retain and forget losses as $\nabla_\theta \mathcal{L}_r(\theta)$ and $\nabla_\theta \mathcal{L}_f(\theta)$, respectively. To quantify their alignment, we use cosine similarity:

$$\mathrm{Sim}(\nabla_\theta \mathcal{L}_f, \nabla_\theta \mathcal{L}_r) = \frac{\langle \nabla_\theta \mathcal{L}_f, \nabla_\theta \mathcal{L}_r \rangle}{\|\nabla_\theta \mathcal{L}_f\| \, \|\nabla_\theta \mathcal{L}_r\|}, \tag{2}$$

which captures directional alignment, abstracting away differences in magnitude. In the context of unlearning, this is crucial: if $\nabla_\theta \mathcal{L}_r(\theta)$ and $\nabla_\theta \mathcal{L}_f(\theta)$ are highly aligned, forgetting updates may interfere with retention, motivating decorrelation.

**Penalty Parameter.** To ensure tractability, we will use a penalty parameter $\rho > 0$, which enforces the stationarity condition of the inner maximization through a soft constraint. This allows us to avoid solving the inner problem to completion while still preserving its structure.

A full notation summary is provided in Appendix 7.1 which will be used throughout the paper.

## 3.2 PROBLEM SETUP

The goal of machine unlearning is to remove the influence of the forget set $\mathcal{D}_f$ from a trained model, while preserving performance on the retain set $\mathcal{D}_r$. Formally, we seek model parameters $\theta$ such that the model's predictions are independent of $\mathcal{D}_f$, yet its performance on $\mathcal{D}_r$ remains optimal.

However, these objectives are often in conflict: naively increasing the loss on $\mathcal{D}_f$ can significantly degrade utility on $\mathcal{D}_r$. To address this, we introduce a bi-level optimization framework that explicitly separates the forgetting and utility objectives. The inner problem seeks to maximize forgetting and decorrelate the influence of $\mathcal{D}_f$ and $\mathcal{D}_r$ via a similarity penalty, while the outer problem restores utility on $\mathcal{D}_r$. The resulting bi-level optimization problem is:

$$\min_{\theta \in \mathbb{R}^d} \quad \mathcal{L}_r(\theta) \qquad \text{subject to} \quad \theta \in \arg\max_{\theta' \in \mathbb{R}^d} \left[ \mathcal{L}_f(\theta') - \beta \cdot \text{Sim}\left(\nabla_\theta \mathcal{L}_f(\theta'), \nabla_\theta \mathcal{L}_r(\theta')\right) \right], \quad (3)$$

where $\beta > 0$ controls the strength of the gradient decorrelation penalty. For ease of presentation, we define

$$\Phi(\theta) := \mathcal{L}_f(\theta) - \beta \cdot \text{Sim}\left(\nabla_\theta \mathcal{L}_f(\theta), \nabla_\theta \mathcal{L}_r(\theta)\right). \quad (4)$$

The formulation in equation 3 ensures that:

- The inner maximization emphasizes forgetting on $\mathcal{D}_f$ while decorrelating from $\mathcal{D}_r$.
- The outer minimization restores utility on $\mathcal{D}_r$, subject to the constraint that forgetting has already been enforced.

## 3.3 BI-LEVEL FORMULATION

We now explicitly separate the two objectives via bi-level optimization.

**Inner Maximization (Forgetting and Decorrelation).** The inner problem seeks parameters that maximize the loss on the forget set while minimizing the similarity between the gradients of the forget and retain losses. This is achieved by solving:

$$\theta^*_{\text{in}} = \arg\max_{\theta'} \left[ \Phi(\theta') \right]. \quad (5)$$

**Outer Minimization (Utility Restoration).** Given the solution $\theta^*_{\text{in}}$ from the inner problem, the outer problem seeks to minimize the loss on the retain set:

$$\theta^* = \arg\min_{\theta \leftarrow \theta^*_{\text{in}}} \mathcal{L}_r(\theta). \quad (6)$$

**Stationarity Constraint.** The bi-level structure enforces that the final model parameters $\theta^*$ are stationary points of the inner maximization objective, i.e., $\nabla_\theta \Phi(\theta^*) = 0$.

## 3.4 PENALTY-BASED SINGLE-LEVEL REFORMULATION

Directly solving the bi-level optimization problem is computationally challenging, especially for large-scale models, due to repeated inner maximization and higher-order derivatives computation. To address this, we adopt a penalty-based single-level reformulation that transforms the bi-level problem into a tractable unconstrained optimization.

**Penalty Reformulation.** We introduce a penalty term that enforces the stationarity condition of the inner maximization as a soft constraint. The resulting objective is:

$$F(\theta) = \mathcal{L}_r(\theta) + \rho \left\| \nabla_\theta \Phi(\theta) \right\|^2, \quad (7)$$

where $\Phi(\theta) = \mathcal{L}_f(\theta) - \beta \cdot \text{Sim}(\nabla_\theta \mathcal{L}_f(\theta), \nabla_\theta \mathcal{L}_r(\theta))$ and $\rho > 0$ penalizes deviation from stationarity. As $\rho$ increases, the penalty term forces $\theta$ to approach the stationary point of the inner objective $\Phi(\theta)$. In the limit as $\rho \to \infty$, any minimizer of $F(\theta)$ satisfies the original bi-level constraint $\nabla_\theta \Phi(\theta) = 0$. This formulation transforms a nested optimization into a tractable single-level objective while preserving the hierarchical structure.

### 3.5 PRACTICAL ALGORITHM AND IMPLEMENTATION

The penalty-based OFMU algorithm is designed for scalability and efficiency in large-scale deep learning settings. Although the penalty reformulation converts the original bi-level problem into a single-level objective, directly optimizing the full loss landscape can be unstable and computationally inefficient. To address this, we adopt a two-loop optimization scheme that alternates between maximizing forgetting and minimizing penalized retain loss. Here, we describe the motivation for this design, its computational advantages, and the implementation details.

**Motivation for Two-Loop Optimization.** While the penalty-based reformulation enables direct optimization of a single objective, the landscape of $F(\theta) = \mathcal{L}_r(\theta) + \rho\|\nabla_\theta\Phi(\theta)\|^2$ can be highly non-convex, especially for deep models. A naive approach may struggle to find good stationary points, especially in the presence of conflicting gradient signals from forgetting and retention. The two-loop scheme mitigates this issue by explicitly maximizing the inner objective $\Phi(\theta)$, which captures both the forget loss and the gradient decorrelation penalty, before each outer update. This design has two key benefits: (i) it encourages the model to traverse regions of the parameter space that are explicitly optimized for forgetting. (ii) it improves stability and convergence by warm-starting each outer iteration from a locally optimized initialization.

Importantly, in the non-convex setting typical of deep learning, the theoretical guarantee is convergence to a stationary point of the penalty objective $F(\theta)$. This point corresponds to a local minimum, maximum, or saddle point. However, because the inner objective $\Phi(\theta)$ is maximized using gradient ascent, the algorithm is biased toward stationary points that are local maxima of the inner objective, which aligns with the unlearning goal. This type of guarantee is the strongest possible in general non-convex optimization and is standard in the literature for deep learning and LLMs.

In contrast, the original bi-level formulation requires fully solving the inner maximization to convergence at each outer iteration, which is computationally prohibitive for large models. Our proposed approach avoids this bottleneck while still enforcing the desired stationarity constraint. For example, in LLMs, a full bi-level update may require thousands of inner steps or even retraining, whereas our approach typically uses a small, fixed $T$ (e.g., $T = 5$ or $10$), dramatically reducing compute cost.

**Two-Loop Optimization Scheme.** At each outer iteration $k$, the algorithm alternates between:

1. **Inner Loop (Forgetting Maximization):** Starting from the current parameters $\theta^{(k)}$, run $T$ steps of gradient ascent on the inner objective $\Phi(\theta)$ to increase the forget loss while decorrelating it from retain gradients:

$$\theta'^{(t+1)} = \theta'^{(t)} + \eta_{\text{in}}\nabla_\theta\Phi(\theta'^{(t)}), \quad t = 0, \dots, T-1, \tag{8}$$

   where $\eta_{\text{in}}$ is the inner learning rate and $\theta'^{(0)} = \theta^{(k)}$. The final inner iterate $\theta_{\text{in}}^{(k)} = \theta'^{(T)}$ serves as the initialization for the outer loop.

2. **Outer Loop (Utility Preservation with Penalty):** The outer step minimizes the retain loss $\mathcal{L}_r(\theta)$ while enforcing the stationarity condition of the inner objective $\Phi(\theta)$. Formally, the update is given by

$$\theta^{(k+1)} = \theta_{\text{in}}^{(k)} - \eta_{\text{out}}\nabla_\theta F(\theta_{\text{in}}^{(k)}), \tag{9}$$

   where $\eta_{\text{out}}$ is the outer learning rate, and $\nabla_\theta F(\theta_{\text{in}}^{(k)}) = \nabla_\theta\mathcal{L}_r(\theta_{\text{in}}^{(k)}) + 2\rho_k\nabla_\theta^2\Phi(\theta_{\text{in}}^{(k)})\nabla_\theta\Phi(\theta_{\text{in}}^{(k)})$.

Here, $\rho_k$ is the penalty parameter at iteration $k$, $\nabla_\theta\Phi$ is the gradient of the inner objective, and $\nabla_\theta^2\Phi$ is its Hessian. The second term, $2\rho_k\nabla_\theta^2\Phi\nabla_\theta\Phi$, results from differentiating the term $\rho_k\|\nabla_\theta\Phi(\theta)\|^2$ with respect to $\theta$, which requires computing a Hessian-vector product (see Appendix 7.7 for details on its efficient computation via automatic differentiation).

**Penalty Schedule and Practical Considerations.** A growing penalty parameter gradually strengthens the enforcement of the inner stationarity condition $\nabla_\theta\Phi(\theta) = 0$. In practice, we adopt an increasing schedule $\rho_{k+1} > \rho_k$: smaller values stabilize the early iterations, while larger values amplify the term $\rho_k\|\nabla_\theta\Phi(\theta)\|^2$, ensuring that violations of stationarity become progressively more costly. A formal justification of this property is provided later in Lemma 1.

**Mini-Batch Stochastic Gradients.** To ensure scalability, all gradients are approximated using mini-batches sampled independently from $\mathcal{D}_f$ and $\mathcal{D}_r$ denoted as $\mathcal{B}_f$ and $\mathcal{B}_r$, each of size $B$. The stochastic gradient estimations $\nabla_\theta\mathcal{L}_f(\theta; \mathcal{B}_f)$ and $\nabla_\theta\mathcal{L}_r(\theta; \mathcal{B}_r)$ reduce the per-iteration computational cost from $O(|\mathcal{D}|)$ to $O(B)$, thereby enabling training on large-scale datasets. While stochasticity introduces variance into the gradient estimates, our two-loop formulation remains robust due

---

**Algorithm 1** Penalty-Based OFMU Bi-Level Unlearning

---

**Require:** Initial parameters $\theta^{(0)}$, penalty schedule $\{\rho_k\}_{k=0}^{K}$, regularization $\beta > 0$, learning rates $\eta_{\text{in}}, \eta_{\text{out}}$, number of outer iterations $K$, number of inner steps $T$, batch size $B$
**Require:** Datasets: $\mathcal{D}_f$ (forget set), $\mathcal{D}_r$ (retain set)
 1: **for** $k = 0, 1, \ldots, K - 1$ **do**
 2:     **(Inner maximization: Forgetting)**
 3:     Initialize $\theta'^{(0)} \leftarrow \theta^{(k)}$
 4:     **for** $t = 0, 1, \ldots, T - 1$ **do**
 5:         Sample mini-batch $\mathcal{B}_f \subset \mathcal{D}_f$, $\mathcal{B}_r \subset \mathcal{D}_r$ of size $B$
 6:         Compute $\nabla_{\theta'} \mathcal{L}_f(\theta'^{(t)}; \mathcal{B}_f)$ and $\nabla_{\theta'} \mathcal{L}_r(\theta'^{(t)}; \mathcal{B}_r)$
 7:         Compute $\text{Sim}(\nabla_\theta \mathcal{L}_f, \nabla_\theta \mathcal{L}_r)$
 8:         $\Phi(\theta'^{(t)}) \leftarrow \mathcal{L}_f(\theta'^{(t)}; \mathcal{B}_f) - \beta \cdot \text{Sim}(\nabla_\theta \mathcal{L}_f, \nabla_\theta \mathcal{L}_r)$
 9:         $\theta'^{(t+1)} \leftarrow \theta'^{(t)} + \eta_{\text{in}} \nabla_{\theta'} \Phi(\theta'^{(t)})$                     ▷ Gradient ascent
10:     **end for**
11:     **(Outer minimization: Utility preservation with penalty)**
12:     Set $\theta_{\text{in}}^{(k)} \leftarrow \theta'^{(T)}$
13:     Sample mini-batch $\mathcal{B}_r' \subset \mathcal{D}_r$ of size $B$
14:     Compute $\nabla_\theta \mathcal{L}_r(\theta_{\text{in}}^{(k)}; \mathcal{B}_r')$
15:     Compute $\nabla_\theta \Phi(\theta_{\text{in}}^{(k)})$ and $\nabla_\theta^2 \Phi(\theta_{\text{in}}^{(k)}) \nabla_\theta \Phi(\theta_{\text{in}}^{(k)})$
16:     $\theta^{(k+1)} \leftarrow \theta_{\text{in}}^{(k)} - \eta_{\text{out}} \left( \nabla_\theta \mathcal{L}_r(\theta_{\text{in}}^{(k)}; \mathcal{B}_r') + 2\rho_k \nabla_\theta^2 \Phi(\theta_{\text{in}}^{(k)}) \nabla_\theta \Phi(\theta_{\text{in}}^{(k)}) \right)$   ▷ Gradient descent
17:     Update penalty: $\rho_{k+1} \leftarrow \text{Increase}(\rho_k)$
18: **end for**
19: **Output:** Final parameters $\theta^{(K)}$

---

to the penalty term $\|\nabla_\theta \Phi(\theta)\|^2$, which regularizes the updates and stabilizes convergence. Algorithm 1 formally presents the complete OFMU process, which terminates after a fixed number of outer iterations or once the stationarity criterion is reached.

## 4 THEORETICAL ANALYSIS

We now provide theoretical analysis for the penalty-based bi-level formulation introduced in Section 3. Our analysis establishes theoretical guarantees for the OFMU algorithm, showing that: (i) the penalty reformulation enforces the stationarity condition on the forgetting objective, (ii) the inner maximization step converges under standard assumptions, and (iii) the full two-loop algorithm converges in both convex and non-convex settings. We further provide a separate analysis of the computational complexity of OFMU in Appendix 7.4. These results provide the theoretical foundation for OFMU and validate its design choices.

**Penalty Reformulation Enforces Stationarity.** We first show that the penalty-based single-level reformulation of the bi-level unlearning problem enforces stationarity of the inner objective, i.e., $\nabla_\theta \Phi(\theta) = 0$, as the penalty parameter $\rho$ increases. This result, presented in following lemma with the proof in Appendix 7.2.1 motivates the use of the penalty method in OFMU.

**Lemma 1** (Stationarity via Penalty Reformulation). *Let $\mathcal{L}_r$ and $\Phi$ be continuously differentiable and bounded below. For any sequence $\{\theta_\rho^*\}$ of minimizers of $F(\theta) = \mathcal{L}_r(\theta) + \rho \|\nabla_\theta \Phi(\theta)\|^2$ with $\rho \to \infty$, every accumulation point $\theta^*$ satisfies $\nabla_\theta \Phi(\theta^*) = 0$.*

**Convergence of the Inner Maximization Step.** Next lemma, with the proof provided in Appendix 7.2.2 analyzes the convergence of the inner maximization loop, showing that gradient ascent, which is used to maximize $\Phi(\theta)$ achieves sublinear convergence in the convex setting.

**Lemma 2** (Convergence Inner Maximization). *Let $\Phi(\theta)$ be convex and differentiable with $L$-Lipschitz continuous gradient. Then, applying $T$ steps of gradient ascent: $\theta'^{(t+1)} = \theta'^{(t)} + \eta_{in} \nabla \Phi(\theta'^{(t)})$ with step size $0 < \eta_{in} \leq 1/L$ yields the bound:*

$$\Phi(\theta_{in}^*) - \Phi(\theta'^{(T)}) \leq \frac{\|\theta_{in}^* - \theta'^{(0)}\|^2}{2T\eta_{in}},$$

*where $\theta_{in}^* = \arg\max_\theta \Phi(\theta)$.*

**Convergence of the Full Two-Loop Algorithm.** Finally, we establish convergence guarantees for the penalty-based OFMU algorithm in both convex and non-convex regimes in the following lemma with the proof provided in Appendix 7.3.

**Lemma 3** (Convergence of Penalty-Based OFMU). *Under Assumptions 7.3, the penalty-based OFMU algorithm converges in both convex and non-convex settings:*

- *Convex case: If $\mathcal{L}_r(\theta)$ and $\Phi(\theta)$ are convex and L-smooth, then after $K$ outer iterations with $T$ inner steps per iteration, the suboptimality satisfies*

$$F(\theta^{(K)}) - F^* \leq \mathcal{O}\left(\tfrac{1}{K}\right) + \mathcal{O}\left(\tfrac{K}{T^2}\right).$$

*Setting $K, T = \mathcal{O}(1/\epsilon)$ ensures $\epsilon$-optimality of the penalty objective.*

- *Non-convex case: If either $\mathcal{L}_r(\theta)$ or $\Phi(\theta)$ is non-convex but L-smooth, then OFMU converges to an $\epsilon$-stationary point of $F(\theta)$, with the expected squared gradient norm bounded by*

$$\min_{k=0,\dots,K-1} \mathbb{E}\|\nabla F(\theta^{(k)})\|^2 \leq \mathcal{O}\left(\tfrac{1}{K}\right) + \mathcal{O}\left(\tfrac{1}{T}\right) + \mathcal{O}(\sigma^2),$$

*where $\sigma^2$ captures the variance of stochastic gradients.*

## 5 EXPERIMENTS

In this section, we describe the experimental setup and evaluate our proposed method, **OFMU**, on both language and vision tasks to assess its effectiveness and generality. Our experiments are designed to examine three key aspects: (i) whether OFMU achieves strong unlearning efficacy while preserving model utility in LLMs; (ii) how OFMU compares against state-of-the-art unlearning baselines across different benchmarks; and (iii) whether OFMU extends effectively to non-language tasks such as vision-based classification.

### 5.1 EXPERIMENTAL SETUP

We conduct all experiments on two NVIDIA H100 80GB GPUs and two NVIDIA H100 NVL 96GB GPUs. For LLMs, we consider two widely used benchmarks: (i) **TOFU** (Maini et al., 2024), a synthetic QA dataset on fictitious authors designed to test entity-level unlearning; (ii) **WMDP** (Li et al., 2024), which evaluates unlearning in high-stakes domains such as biosecurity, cybersecurity, and chemical safety. For vision tasks, we use **CIFAR-10** and **CIFAR-100** (Krizhevsky & Hinton, 2009) and evaluate OFMU under two settings: (i) class-wise forgetting, where all examples from one or more classes are removed and (ii) random forgetting, where a randomly selected subset spanning all classes is removed. Sections 5.2 and 5.3 present results on the **TOFU** and **CIFAR-10** benchmarks respectively, while results for **WMDP** and **CIFAR-100**, together with complete details of the experimental setup, evaluation metrics, baselines, and models, are deferred to Appendix 7.5.

### 5.2 TOFU RESULTS

For TOFU benchmark, we evaluate OFMU across three forgetting scenarios: `forget01`, `forget05`, and `forget10`, which correspond to removing 1%, 5%, and 10% of the dataset, respectively, using two model architectures: `LLaMA-2-7B-hf-chat`[1] and `LLaMA-3.2-1B-Instruct`[2]. We report performance using three key metrics: forget quality (FQ), model utility (MU), and forget truth ratio (FTR), where higher values indicate more effective forgetting, better utility retention, and stronger reliability of unlearned outputs, respectively.

**Forgetting Quality.** On both architectures, OFMU achieves strong FQ, comparable to or exceeding preference-based methods such as NPO (Bourtoule et al., 2021). In `forget01`, OFMU achieves slightly lower FQ than NPO on LLaMA-2 but matches or surpasses it on LLaMA-3.2, while maintaining stronger MU and FTR. This highlights that our framework prioritizes balance rather than over-optimizing a single metric. Unlike Gradient Ascent (GA) (Thudi et al., 2022) and Gradient Difference (GD) (Maini et al., 2024), which aggressively maximize the forget loss but collapse to near-zero utility, OFMU enforces forgetting without destabilizing updates.

**Utility Preservation.** MU is where many baselines diverge. Methods like RMU (Li et al., 2024) preserve utility well but at the cost of incomplete forgetting, while GA achieves near-perfect forgetting

---

[1] https://huggingface.co/meta-llama/Llama-2-7b-chat-hf
[2] https://huggingface.co/open-unlearning/tofu_Llama-3.2-1B-Instruct_full

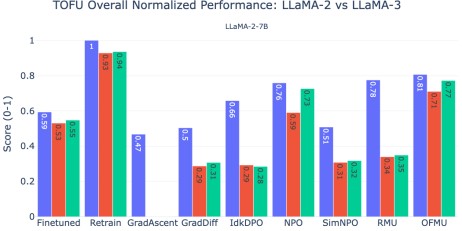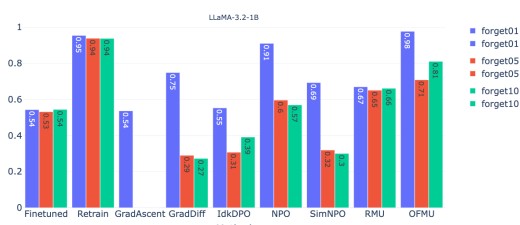

Figure 2: Overall normalized performance of unlearning methods on LLaMA-2 and LLaMA-3 under different forget scenarios (1%, 5%, 10%). The overall score is computed by normalizing FQ, MU, and FTR and then averaging them. Higher scores indicate better balance between forgetting efficacy and utility preservation. Details about the calculation are provided in Appendix 7.5.3

but eliminates MU entirely. In `forget05`, although GA attains higher raw FQ, its utility completely collapses (MU = 0.00), whereas OFMU sustains competitive FQ with substantially higher MU (0.65 on LLaMA-2). Similarly, in `forget10`, OFMU preserves robustness across all metrics, while GA and GD sacrifice utility entirely, and NPO degrades sharply. Overall, OFMU maintains MU close to the Retain baseline while still enforcing strong forgetting.

**Truth Ratio.** FTR further confirms OFMU's balanced behavior. Whereas GA and GD degrade truthfulness due to unstable optimization, and NPO variants sometimes inflate FTR by overfitting to retain data. OFMU consistently maintains high FTR across all scenarios. This indicates that unlearned models continue to provide reliable responses, rather than memorized or distorted ones.

Table 1: Performance of unlearning methods on TOFU using `LLaMA-2-7B-hf-chat` (mean ± std over 5 runs).

| Method | forget01 | | | forget05 | | | forget10 | | |
|---|---|---|---|---|---|---|---|---|---|
| | FQ↑ | MU↑ | FTR↑ | FQ↑ | MU↑ | FTR↑ | FQ↑ | MU↑ | FTR↑ |
| Finetuned | 1.32±0.08e-3 | 0.64±0.02 | 0.56±0.03 | 5.68±0.14e-14 | 0.63±0.01 | 0.50±0.02 | 4.41±0.11e-25 | 0.64±0.01 | 0.54±0.02 |
| Retrain | 1.00±0.00 | 0.63±0.01 | 0.70±0.02 | 1.00±0.00 | 0.63±0.01 | 0.67±0.02 | 1.00±0.00 | 0.62±0.02 | 0.69±0.02 |
| GradAscent | 1.91±0.10e-4 | 0.55±0.03 | 0.37±0.04 | 1.84±0.07e-119 | 0.00±0.00 | 8.71±0.19e-96 | 1.12±0.06e-239 | 0.00±0.00 | 2.16±0.11e-32 |
| GradDiff | 3.14±0.14e-3 | 0.57±0.02 | 0.42±0.03 | 2.01±0.09e-119 | 0.59±0.03 | 4.20±0.17e-95 | 1.86±0.08e-229 | 0.58±0.02 | 1.49±0.06e-7 |
| IdkDPO | 0.12±0.02 | 0.57±0.03 | 0.68±0.02 | 4.00±0.20e-6 | 0.04±0.01 | 0.67±0.02 | 5.40±0.25e-13 | 0.04±0.01 | 0.64±0.03 |
| NPO | 0.40±0.04 | 0.58±0.02 | 0.64±0.02 | 0.09±0.02 | 0.53±0.03 | 0.71±0.02 | 0.42±0.04 | 0.54±0.02 | 0.74±0.02 |
| SimNPO | 1.31±0.09e-3 | 0.58±0.02 | 0.41±0.03 | 1.10±0.05e-106 | 0.60±0.02 | 3.88±0.17e-5 | 1.52±0.08e-198 | 0.60±0.01 | 3.10±0.15e-4 |
| RMU | 0.41±0.04 | 0.62±0.01 | 0.65±0.02 | 9.61±0.24e-10 | 0.02±0.01 | 0.81±0.02 | 6.98±0.21e-21 | 0.03±0.01 | 0.82±0.02 |
| **OFMU (ours)** | **0.44±0.03** | **0.63±0.01** | **0.68±0.02** | **0.14±0.02** | **0.65±0.02** | **0.82±0.01** | **0.42±0.03** | **0.61±0.02** | **0.77±0.02** |

To capture a holistic view of unlearning efficacy, we aggregate the three core metrics — FQ, MU, FTR into a single normalized score (Figure 2). This unified view highlights the balance between forgetting and retention across different forget scenarios and shows how OFMU strikes the balance to achieve overall better results in unlearning.

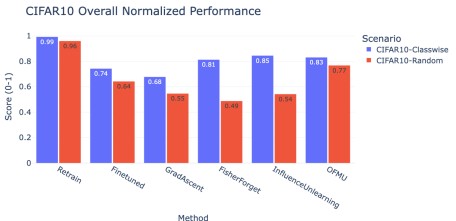

Figure 3: Overall normalized performance of unlearning methods on CIFAR-10. The score is obtained by normalizing four key metrics—UA, RA, TA, and MIA Efficacy— within each scenario and averaging them into a unified value. Details of the calculation are given in Appendix 7.5.3.

## 5.3 CIFAR-10 RESULTS

For CIFAR-10, we evaluate OFMU under two settings: *class-wise forgetting*, where an entire class is removed, and *random forgetting*, where 10% of the training data is randomly selected as the forget set. The results are summarized in Table 3. We report Unlearning Accuracy (UA), Retain Accuracy (RA), Total Accuracy (TA), and Membership Inference Attack efficacy (MIA-Efficacy), where higher values indicate better unlearning performance and robustness.

**Class-wise Forgetting.** Retraining from scratch achieves perfect unlearning (100% UA) and strong overall utility (94.80% RA, 91.82% TA), but is com-

putationally infeasible. Among approximate methods, Fisher Forget (FF) (Golatkar et al., 2021) and Influence Unlearning (IU) (Mehta et al., 2022) preserve utility reasonably well, with IU in particular showing the highest UA (89.31%). However, IU is computationally expensive, requiring repeated influence function estimations and parameter adjustments, which makes it impractical for LLMs with billions of parameters. In contrast, OFMU achieves a balanced trade-off: 81.51% UA, 93.51% RA, and 86.88% TA. While its UA is slightly lower than IU, OFMU generalizes more effectively, as shown by its higher MIA-Efficacy (59.76). This indicates that OFMU not only forgets the targeted class but also improves robustness against membership inference attacks, a critical security measure. Unlike IU, OFMU scales naturally to deep non-convex settings without prohibitive computational cost, making it more suitable for practical deployment in LLMs.

Table 2: Performance of unlearning methods on TOFU using `LLaMA-3.2-1B-Instruct` (mean $\pm$ std over 5 runs).

| Method | forget01 | | | forget05 | | | forget10 | | |
| | FQ↑ | MU↑ | FTR↑ | FQ↑ | MU↑ | FTR↑ | FQ↑ | MU↑ | FTR↑ |
|---|---|---|---|---|---|---|---|---|---|
| Finetuned | 0.011±0.003 | 0.60±0.02 | 0.48±0.03 | 1.28±0.06e-13 | 0.60±0.01 | 0.48±0.02 | 1.69±0.07e-21 | 0.60±0.01 | 0.49±0.02 |
| Retrain | 1.00±0.00 | 0.60±0.02 | 0.66±0.02 | 1.00±0.00 | 0.60±0.01 | 0.65±0.02 | 1.00±0.00 | 0.59±0.02 | 0.64±0.02 |
| GradAscent | 0.28±0.04 | 0.35±0.04 | 0.58±0.03 | 1.88±0.08e-119 | 0.00±0.00 | 2.47±0.12e-23 | 1.09±0.05e-239 | 0.00±0.00 | 2.20±0.11e-18 |
| GradDiff | 0.78±0.05 | 0.44±0.03 | 0.58±0.03 | 1.95±0.09e-119 | 0.54±0.03 | 3.92±0.17e-34 | 1.10±0.05e-239 | 0.50±0.03 | 3.57±0.18e-27 |
| IdkDPO | 0.01±0.003 | 0.51±0.03 | 0.60±0.03 | 1.10±0.05e-5 | 0.07±0.02 | 0.62±0.03 | 4.60±0.22e-12 | 0.23±0.03 | 0.60±0.03 |
| NPO | 0.92±0.04 | 0.57±0.02 | 0.66±0.02 | 0.14±0.02 | 0.46±0.03 | 0.70±0.02 | 0.02±0.005 | 0.46±0.03 | 0.70±0.02 |
| SimNPO | 0.58±0.04 | 0.46±0.03 | 0.56±0.03 | 5.00±0.25e-100 | 0.58±0.02 | 4.18±0.20e-3 | 2.45±0.12e-203 | 0.54±0.02 | 1.06±0.05e-5 |
| RMU | 0.17±0.03 | 0.56±0.02 | 0.71±0.02 | 4.91±0.23e-10 | 0.59±0.02 | 0.78±0.02 | 3.19±0.14e-15 | 0.59±0.02 | 0.77±0.02 |
| **OFMU (ours)** | **0.91±0.04** | **0.61±0.02** | **0.75±0.02** | **0.15±0.02** | **0.61±0.02** | **0.75±0.02** | **0.41±0.03** | **0.60±0.02** | **0.77±0.02** |

**Random Forgetting.** The random forgetting task is more challenging, since the forget set is dispersed rather than concentrated. Retraining achieves the highest UA (6.79%), while most approximate methods collapse, with UA close to zero (e.g., GA: 0.78%, FF: 0.51%). These methods struggle to generalize forgetting uniformly across the randomly distributed forget samples, highlighting their sensitivity to the structure of the forget set. OFMU achieves 7.71% UA, slightly outperforming retraining. While its RA (92.25%) and TA (88.61%) are marginally lower than retrain or fine-tuning, OFMU maintains a better balance by reducing susceptibility to membership inference (MIA-Efficacy: 3.36 versus 1.21 and 1.87 for FF and GA, respectively). This shows that OFMU enforces forgetting more uniformly, even in the scattered random setting, without collapsing utility.

Table 3: Performance of unlearning methods on CIFAR-10 under **class-wise** and **random** forgetting. Values show mean $\pm$ standard deviation over 5 runs.

| Method | Class-wise Forgetting | | | | Random Forgetting (10% forget set) | | | |
| | UA ↑ | RA ↑ | TA ↑ | MIA ↑ | UA ↑ | RA ↑ | TA ↑ | MIA ↑ |
|---|---|---|---|---|---|---|---|---|
| Retrain | 100.00 ± 0.0 | 94.80 ± 0.2 | 91.82 ± 0.3 | 100.00 ± 0.0 | 6.79 ± 0.3 | 100.00 ± 0.0 | 92.04 ± 0.1 | 16.08 ± 0.5 |
| Finetuned (FT) | 42.43 ± 2.1 | 94.19 ± 0.5 | 94.61 ± 0.6 | 56.51 ± 2.8 | 1.82 ± 0.2 | 99.54 ± 0.2 | 92.84 ± 0.4 | 5.66 ± 0.4 |
| GradAscent (GA) | 37.11 ± 2.2 | 86.52 ± 1.7 | 82.41 ± 2.0 | 55.03 ± 2.7 | 0.78 ± 0.3 | 99.38 ± 0.3 | 92.10 ± 0.7 | 1.87 ± 0.4 |
| Fisher Forget (FF) | 79.71 ± 1.4 | 94.12 ± 0.4 | 93.96 ± 0.6 | 46.38 ± 2.4 | 0.51 ± 0.2 | 88.03 ± 1.8 | 87.70 ± 1.9 | 1.21 ± 0.4 |
| Influence Unlearning (IU) | 89.31 ± 1.1 | 92.19 ± 0.7 | 90.63 ± 1.0 | 55.22 ± 2.5 | 0.62 ± 0.3 | 99.39 ± 0.3 | 94.43 ± 0.6 | 1.51 ± 0.3 |
| **OFMU (ours)** | **81.51 ± 1.3** | **93.51 ± 0.6** | **86.88 ± 1.2** | **59.76 ± 2.4** | **7.71 ± 0.4** | **92.25 ± 0.9** | **88.61 ± 1.1** | **3.36 ± 0.6** |

The CIFAR-10 results further underscore the robustness of OFMU. Whereas existing baselines overemphasize either unlearning or retention, OFMU achieves stable performance across both scenarios, validating the advantages of its hierarchical optimization design. As shown in Figure 3, we also report the overall normalized score for CIFAR-10, analogous to the TOFU benchmark. In the class-wise setting, Influence Unlearning (IU) attains higher unlearning accuracy, but its heavy computational cost limits practicality. In the more challenging random forgetting scenario, where most baselines collapse, OFMU achieves the best overall performance. These results confirm that OFMU maintains a consistent balance across metrics, achieving effective and robust unlearning without sacrificing generalization, unlike methods that over-optimize for a single objective.

## 6 CONCLUSION AND FUTURE WORK

In this work, we introduced **OFMU**, a penalty-based bi-level framework for machine unlearning that explicitly prioritizes forgetting before utility preservation. By combining a scalable two-loop algorithm with a similarity-aware penalty, OFMU achieves state-of-the-art trade-offs across language and vision benchmarks. Our theoretical analysis provides convergence guarantees in convex and non-convex regimes, and empirical results consistently demonstrate improved stability, robustness

to hard-to-forget samples, and stronger resilience against membership inference attacks compared to existing approaches.

Although OFMU makes significant progress, several directions remain open. First, extending OFMU to continual unlearning scenarios, where multiple requests arrive sequentially, would enhance its applicability. Second, investigating adaptive penalty schedules and alternative gradient similarity measures could further improve robustness. Finally, applying OFMU to even larger foundation models and diverse modalities such as speech and multimodal learning presents a promising avenue for future research.

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

# 7 APPENDIX

## 7.1 NOTATION SUMMARY

- $\theta \in \mathbb{R}^d$: model parameters.
- $\mathcal{D} = \mathcal{D}_r \cup \mathcal{D}_f$: full dataset.
- $\mathcal{D}_r$: retain set (data to preserve).
- $\mathcal{D}_f$: forget set (data to unlearn).
- $\ell(\cdot, \cdot)$: base loss function (e.g., cross-entropy).
- $\mathcal{L}_r(\theta)$: empirical loss on the retain set.
- $\mathcal{L}_f(\theta)$: empirical loss on the forget set.
- $\Phi(\theta) = \mathcal{L}_f(\theta) - \beta \cdot \mathrm{Sim}(g_f, g_r)$: inner maximization objective.
- $F(\theta) = \mathcal{L}_r(\theta) + \rho\|\nabla_\theta \Phi(\theta)\|^2$: penalty-based reformulated objective.
- $\nabla_\theta \mathcal{L}_r(\theta)$: gradient of the retain loss.
- $\nabla_\theta \mathcal{L}_f(\theta)$: gradient of the forget loss.
- $\mathrm{Sim}(\nabla_\theta \mathcal{L}_f(\theta), \nabla_\theta \mathcal{L}_r(\theta))$: cosine similarity between $\nabla_\theta \mathcal{L}_f(\theta)$ and $\nabla_\theta \mathcal{L}_r(\theta)$.
- $\nabla_\theta^2 \Phi(\theta)$: Hessian of the inner objective.
- $Hv$: Hessian–vector product (Pearlmutter trick).
- $\beta > 0$: regularization parameter for similarity-aware decorrelation.
- $\rho > 0$: penalty parameter enforcing stationarity.
- $\eta_{\mathrm{in}}$: learning rate for the inner loop.
- $\eta_{\mathrm{out}}$: learning rate for the outer loop.
- $T$: number of inner loop steps per outer iteration.
- $K$: number of outer iterations.
- $B$: mini-batch size.
- $\theta_{\mathrm{in}}^{(k)}$: parameters after inner loop at iteration $k$.
- $\theta^{(k)}$: parameters after outer loop at iteration $k$.
- FQ: Forget Quality (lower residual accuracy on forget set).
- MU: Model Utility (performance on retain set).
- FTR: Forget Truth Ratio (faithfulness of unlearning).
- UA: Unlearning Accuracy (CIFAR evaluation).
- RA: Retain Accuracy.
- TA: Total Accuracy.
- MIA-Efficacy: Membership Inference Attack efficacy.
- UDI($x$): Unlearning Difficulty Index for sample $x$.
- $\|\nabla_\theta \mathcal{L}_{\mathrm{forget}}(x)\|_2$: gradient norm of forget loss on $x$.
- $\Delta\ell(x)$: loss margin to the target threshold (used in UDI).
- $\tau(\cdot, \cdot)$: Spearman correlation coefficient.

## 7.2 LEMMA PROOFS

### 7.2.1 PROOF OF LEMMA 1

*Proof.* Let $\theta_\rho^*$ be a minimizer of $F(\theta) = \mathcal{L}_r(\theta) + \rho\|\nabla_\theta \Phi(\theta)\|^2$ for a given $\rho > 0$. Assume $\mathcal{L}_r$ and $\Phi$ are continuously differentiable and bounded below.

Suppose, for contradiction, that there exists an accumulation point $\theta^*$ of the sequence $\{\theta_\rho^*\}$ as $\rho \to \infty$ such that $\nabla_\theta \Phi(\theta^*) \neq 0$. Then, for sufficiently large $\rho$, the penalty term $\rho\|\nabla_\theta \Phi(\theta_\rho^*)\|^2$ would dominate $F(\theta_\rho^*)$, causing it to diverge to infinity, which contradicts the assumption that $F(\theta_\rho^*)$ is minimized and bounded below.

Therefore, it must be that $\nabla_\theta \Phi(\theta^*) = 0$ for any accumulation point $\theta^*$ of the minimizers as $\rho \to \infty$. $\qquad\square$

### 7.2.2 PROOF OF LEMMA 2

*Proof.* Let $d^{(t)} := \theta_{\text{in}}^* - \theta'^{(t)}$. By convexity of $\Phi$, for any $\theta$ and $\theta^*$,

$$\Phi(\theta^*) \leq \Phi(\theta) + \langle \nabla\Phi(\theta), \theta^* - \theta \rangle, \tag{10}$$

so

$$\Phi(\theta_{\text{in}}^*) - \Phi(\theta'^{(t)}) \leq \langle \nabla\Phi(\theta'^{(t)}), d^{(t)} \rangle. \tag{11}$$

The update rule gives:

$$d^{(t+1)} = d^{(t)} - \eta_{\text{in}} \nabla\Phi(\theta'^{(t)}), \tag{12}$$

so

$$\|d^{(t+1)}\|^2 = \|d^{(t)}\|^2 - 2\eta_{\text{in}} \langle d^{(t)}, \nabla\Phi(\theta'^{(t)}) \rangle + \eta_{\text{in}}^2 \|\nabla\Phi(\theta'^{(t)})\|^2. \tag{13}$$

Using the convexity bound above, we have:

$$\langle d^{(t)}, \nabla\Phi(\theta'^{(t)}) \rangle \geq \Phi(\theta_{\text{in}}^*) - \Phi(\theta'^{(t)}). \tag{14}$$

For $L$-smooth convex functions, it yields:

$$\|\nabla\Phi(\theta)\|^2 \leq 2L(\Phi(\theta_{\text{in}}^*) - \Phi(\theta)). \tag{15}$$

Substituting the two bounds from equation 14 and equation 15 into equation 13, we have:

$$\|d^{(t+1)}\|^2 \leq \|d^{(t)}\|^2 - 2\eta_{\text{in}}\big(\Phi(\theta_{\text{in}}^*) - \Phi(\theta'^{(t)})\big) + 2\eta_{\text{in}}^2 L\big(\Phi(\theta_{\text{in}}^*) - \Phi(\theta'^{(t)})\big). \tag{16}$$

This simplifies to

$$\|d^{(t+1)}\|^2 \leq \|d^{(t)}\|^2 - 2\eta_{\text{in}}(1 - L\eta_{\text{in}})(\Phi(\theta_{\text{in}}^*) - \Phi(\theta'^{(t)})). \tag{17}$$

Summing over $t = 0$ to $T - 1$ yields:

$$\|d^{(T)}\|^2 \leq \|d^{(0)}\|^2 - 2\eta_{\text{in}}(1 - L\eta_{\text{in}}) \sum_{t=0}^{T-1} (\Phi(\theta_{\text{in}}^*) - \Phi(\theta'^{(t)})). \tag{18}$$

Since $\|d^{(T)}\|^2 \geq 0$, we obtain:

$$\sum_{t=0}^{T-1} (\Phi(\theta_{\text{in}}^*) - \Phi(\theta'^{(t)})) \leq \frac{\|d^{(0)}\|^2}{2\eta_{\text{in}}(1 - L\eta_{\text{in}})}. \tag{19}$$

Thus, the average suboptimality is give by:

$$\min_t (\Phi(\theta_{\text{in}}^*) - \Phi(\theta'^{(t)})) \leq \frac{\|\theta'^{(0)} - \theta_{\text{in}}^*\|^2}{2\eta_{\text{in}}(1 - L\eta_{\text{in}})T}. \tag{20}$$

For $\eta_{\text{in}} \leq 1/L$, $1 - L\eta_{\text{in}} \geq 1/2$, we achieve:

$$\Phi(\theta_{\text{in}}^*) - \Phi(\theta'^{(T)}) \leq \frac{\|\theta'^{(0)} - \theta_{\text{in}}^*\|^2}{2\eta_{\text{in}}T}. \tag{21}$$

$\square$

### 7.3 CONVERGENCE GUARANTEES FOR PENALTY-BASED OFMU

In this section, we rigorously analyze the convergence properties of the penalty-based OFMU algorithm for bi-level unlearning. We consider both convex and non-convex settings, reflecting the diversity of loss landscapes encountered in practice. Our analysis is grounded in the following problem setup and assumptions.

**Problem Setup.** We study the optimization of the penalty-based objective

$$F(\theta) = \mathcal{L}_r(\theta) + \rho\|\nabla_\theta\Phi(\theta)\|^2, \tag{22}$$

where $\mathcal{L}_r(\theta)$ is the retain loss, $\Phi(\theta)$ is the inner (forgetting) objective, and $\rho > 0$ is the penalty parameter. The algorithm alternates between $T$ steps of gradient ascent on $\Phi(\theta)$ (inner loop) and a single gradient descent step on $F(\theta)$ (outer loop).

**Assumptions.**   Throughout our analysis, we assume:

- $\mathcal{L}_r(\theta)$ and $\Phi(\theta)$ are continuously differentiable.
- The gradients $\nabla_\theta \mathcal{L}_r(\theta)$ and $\nabla_\theta \Phi(\theta)$ are $L$-Lipschitz continuous.
- The penalty parameter $\rho$ is non-decreasing and bounded below by $\rho_{\min} > 0$.
- The inner and outer step sizes satisfy $\eta_{\text{in}} \leq 1/L$ and $\eta_{\text{out}} \leq 1/L_F$, where $L_F$ is the Lipschitz constant of $\nabla F(\theta)$.

Additional assumptions specific to the convex or non-convex setting will be stated in the corresponding subsections.

We now present detailed convergence analyses for both the convex and non-convex cases.

### 7.3.1   CONVERGENCE ANALYSIS: CONVEX CASE

We first analyze the convergence of the penalty-based OFMU algorithm under the assumption that both the retain loss $\mathcal{L}_r(\theta)$ and the inner objective $\Phi(\theta)$ are convex and $L$-smooth. Our goal is to rigorously bound the suboptimality of the penalty-based objective $F(\theta)$ after $K$ outer iterations, each involving $T$ steps of gradient ascent on $\Phi(\theta)$.

**Additional Assumptions (Convex Case).**

- $\mathcal{L}_r(\theta)$ and $\Phi(\theta)$ are convex.

**Algorithmic Steps.**   At each outer iteration $k$:

1. **Inner maximization:** Starting from $\theta^{(k)}$, perform $T$ steps of gradient ascent on $\Phi(\theta)$ with step size $\eta_{\text{in}} \leq 1/L$ to obtain $\theta_{\text{in}}^{(k)}$.
2. **Outer minimization:** Update $\theta^{(k+1)} = \theta_{\text{in}}^{(k)} - \eta_{\text{out}} \nabla F(\theta_{\text{in}}^{(k)})$, where $\eta_{\text{out}} \leq 1/L_F$.

**Step 1: Inner Maximization Error.**   By Lemma 2, after $T$ steps of gradient ascent on the convex, $L$-smooth function $\Phi$, we have

$$\Phi(\theta_{\text{in}}^*) - \Phi(\theta_{\text{in}}^{(k)}) \leq \frac{\|\theta_{\text{in}}^* - \theta^{(k)}\|^2}{2T\eta_{\text{in}}}, \tag{23}$$

where $\theta_{\text{in}}^* = \arg\max_\theta \Phi(\theta)$. This quantifies the inexactness of the inner maximization.

**Step 2: Outer Minimization with Inexact Inner Solution.**   The outer update is performed using $\theta_{\text{in}}^{(k)}$ as input. Since $F(\theta)$ is convex and $L_F$-smooth, the standard inexact gradient descent analysis yields:

$$F(\theta^{(k+1)}) \leq F(\theta_{\text{in}}^{(k)}) - \frac{\eta_{\text{out}}}{2}\|\nabla F(\theta_{\text{in}}^{(k)})\|^2. \tag{24}$$

Summing over $k = 0$ to $K - 1$ and rearranging, we obtain

$$\frac{1}{K}\sum_{k=0}^{K-1}\|\nabla F(\theta_{\text{in}}^{(k)})\|^2 \leq \frac{2(F(\theta^{(0)}) - F^*)}{K\eta_{\text{out}}}, \tag{25}$$

where $F^*$ is the minimum value of $F$.

**Step 3: Bounding the Total Suboptimality.**   Due to the inexactness of the inner maximization, the update direction is not the true minimizer of the inner problem. The error in the outer update can be bounded in terms of the inner error. Specifically, the gradient error at each step is

$$\delta^{(k)} := \nabla F(\theta_{\text{in}}^{(k)}) - \nabla F(\theta^{(k)}), \tag{26}$$

and, by smoothness, $\|\delta^{(k)}\| \leq L_F \|\theta_{\text{in}}^{(k)} - \theta^{(k)}\|$. Since $\|\theta_{\text{in}}^{(k)} - \theta^{(k)}\|$ is controlled by the inner maximization error, and by Lemma 2 this error is $\mathcal{O}(1/T)$, we have $\|\delta^{(k)}\|^2 = \mathcal{O}(1/T^2)$. Thus, the cumulative error over $K$ steps scales as $\mathcal{O}(K/T^2)$.

**Step 4: Final Rate and Parameter Choices.** Combining the above, the suboptimality after $K$ iterations is bounded by

$$F(\theta^{(K)}) - F^* \leq \frac{\|\theta^{(0)} - \theta^*\|^2}{2K\eta_{\text{out}}} + \mathcal{O}\left(\frac{K}{T^2}\right). \tag{27}$$

To achieve $\mathcal{O}(\epsilon)$ suboptimality, it suffices to choose $K = \mathcal{O}(1/\epsilon)$ and $T = \mathcal{O}(1/\epsilon)$.

The penalty-based OFMU algorithm, under convexity and smoothness assumptions, converges to an $\epsilon$-optimal solution of the penalty objective at a sublinear rate, with explicit dependence on the number of outer and inner iterations. The analysis leverages Lemma 1 for stationarity enforcement and Lemma 2 for the inner maximization rate.

### 7.3.2 CONVERGENCE ANALYSIS: NON-CONVEX CASE

We now analyze the convergence of the penalty-based OFMU algorithm in the non-convex setting, where either $\mathcal{L}_r(\theta)$ or $\Phi(\theta)$ (or both) may be non-convex. In this regime, global optimality is generally intractable, so our goal is to establish convergence to an $\epsilon$-stationary point of the penalty objective $F(\theta)$.

**Assumptions (Non-Convex Case).**

- $\mathcal{L}_r(\theta)$ and $\Phi(\theta)$ are differentiable (possibly non-convex) and $L$-smooth.
- The stochastic gradients used in the inner loop are unbiased with bounded variance $\sigma^2$.
- The penalty objective $F(\theta)$ is $L_F$-smooth and lower bounded.
- $\|\nabla\mathcal{L}_r(\theta)\| \leq G_r$ and $\|\nabla^2\Phi(\theta)\| \leq H$ for all $\theta$.

**Step 1: Inner Loop Approximation.** By standard results for stochastic gradient ascent on $L$-smooth non-convex functions (see, e.g., (Ghadimi & Lan, 2013)), after $T$ steps we have

$$\frac{1}{T}\sum_{t=0}^{T-1}\mathbb{E}\left[\|\nabla\Phi(\theta'^{(t)})\|^2\right] \leq \frac{2(\Phi^* - \Phi(\theta'^{(0)}))}{\eta_{\text{in}}T} + \eta_{\text{in}}L\sigma^2, \tag{28}$$

where $\Phi^*$ is the maximum value of $\Phi$. Thus, the expected squared gradient norm at the final inner iterate satisfies

$$\mathbb{E}[\|\nabla\Phi(\theta_{\text{in}}^{(k)})\|^2] \leq \mathcal{O}\left(\frac{1}{T}\right) + \mathcal{O}(\sigma^2). \tag{29}$$

**Step 2: Outer Loop Descent and Stationarity.** The outer update is $\theta^{(k+1)} = \theta_{\text{in}}^{(k)} - \eta_{\text{out}}\nabla F(\theta_{\text{in}}^{(k)})$. Since $F$ is $L_F$-smooth, the standard descent lemma gives

$$F(\theta^{(k+1)}) \leq F(\theta_{\text{in}}^{(k)}) - \frac{\eta_{\text{out}}}{2}\|\nabla F(\theta_{\text{in}}^{(k)})\|^2. \tag{30}$$

Summing over $K$ iterations and rearranging, we obtain

$$\frac{1}{K}\sum_{k=0}^{K-1}\mathbb{E}\left[\|\nabla F(\theta_{\text{in}}^{(k)})\|^2\right] \leq \frac{2(F(\theta^{(0)}) - F^*)}{K\eta_{\text{out}}}. \tag{31}$$

**Step 3: Explicit Dependence on Inner Loop Error.** The gradient of the penalty objective is

$$\nabla F(\theta) = \nabla\mathcal{L}_r(\theta) + 2\rho\nabla^2\Phi(\theta)\nabla\Phi(\theta). \tag{32}$$

Using the inequality $\|\mathbf{a} + \mathbf{b}\|^2 \leq 2\|\mathbf{a}\|^2 + 2\|\mathbf{b}\|^2$ and assuming $\|\nabla\mathcal{L}_r(\theta)\| \leq G_r$ and $\|\nabla^2\Phi(\theta)\| \leq H$, we have

$$\mathbb{E}\left[\|\nabla F(\theta_{\text{in}}^{(k)})\|^2\right] \leq 2G_r^2 + 8\rho^2 H^2 \cdot \mathbb{E}\left[\|\nabla\Phi(\theta_{\text{in}}^{(k)})\|^2\right]. \tag{33}$$

Plugging in the bound from the inner loop,

$$\mathbb{E}\left[\|\nabla F(\theta_{\text{in}}^{(k)})\|^2\right] \leq 2G_r^2 + 8\rho^2 H^2\left(\frac{2(\Phi^* - \Phi(\theta'^{(0)}))}{\eta_{\text{in}}T} + \eta_{\text{in}}L\sigma^2\right). \tag{34}$$

**Step 4: Final Convergence Guarantee.** Combining the outer-loop descent bound for K steps and the inner approximation error, we obtain

$$\min_k \mathbb{E}\|\nabla F(\theta_{\text{in}}^{(k)})\|^2 \leq \frac{2(F(\theta^{(0)}) - F^*)}{K\eta_{\text{out}}} + 2G_r^2 + 8\rho^2 H^2 \left( \frac{2(\Phi^* - \Phi(\theta'^{(0)}))}{\eta_{\text{in}} T} + \eta_{\text{in}} L\sigma^2 \right) \quad (35)$$

This result shows that the penalty-based OFMU algorithm converges to an $\epsilon$-stationary point of the penalty objective. The convergence rate exhibits explicit dependence on the number of outer iterations $K$, inner-loop accuracy $T$, penalty parameter $\rho$, curvature bound $H$, and gradient noise variance $\sigma^2$. Choosing $K = \mathcal{O}(1/\epsilon)$ and $T = \mathcal{O}(1/\epsilon)$ ensures convergence to an $\epsilon$-stationary point in expectation.

## 7.4 COMPUTATIONAL COMPLEXITY AND PRACTICAL EFFICIENCY

This section provides a formal yet practical analysis of the computational cost of OFMU and explains why the method remains scalable despite introducing a bi-level structure. In particular, we highlight two design choices—(i) the small number of inner steps $T$, and (ii) the use of Hessian–vector products (HVPs)—that keep our overall cost close to standard gradient based single-loop optimization while still enforcing the hierarchical structure introduced in Section 3.

### 7.4.1 PER-ITERATION COST OF THE INNER AND OUTER LOOPS

Let $d$ denote the number of trainable parameters, $B$ the minibatch size, and $(K, T)$ the number of outer and inner iterations, respectively. A standard forward–backward computation on a minibatch incurs

$$C_{\text{fb}} = \Theta(Bd)$$

floating-point operations. This is the dominant unit of cost in both the inner and outer loops.

**Inner loop cost.** Each inner step requires one gradient of $\Phi(\theta)$, which itself evaluates $\nabla\mathcal{L}_f, \nabla\mathcal{L}_r$, and their cosine similarity. All of these operations share the same forward–backward structure, so each inner step costs $\Theta(Bd)$, yielding

$$C_{\text{inner}} = \Theta(TBd).$$

**Outer loop cost.** The outer step requires: (i) a gradient of $\mathcal{L}_r(\theta)$ and (ii) one HVP term $\nabla^2\Phi(\theta)\nabla\Phi(\theta)$ arising from the penalty objective (Section 3). The HVP is computed via a Pearlmutter's method (Appendix 7.7) and therefore costs *the same order* as a gradient with detail given in next section:

$$C_{\text{HVP}} = \Theta(Bd).$$

Thus the outer step costs

$$C_{\text{outer}} = \Theta(Bd).$$

### 7.4.2 WHY HESSIAN–VECTOR PRODUCTS ARE COMPUTATIONALLY CHEAP

A naïve Hessian computation requires $O(d^2)$ memory and time, which is infeasible for modern LLMs. However, OFMU uses the Hessian only through the vector product

$$\nabla^2\Phi(\theta)\,v,$$

which can be implemented using Pearlmutter's method (Appendix 7.7). This trick computes $Hv$ using *one additional backward pass* without forming the matrix. Consequently,

$$C_{\text{HVP}} \approx C_{\text{grad}}$$

up to a small constant factor. Therefore, the HVP introduces negligible overhead compared to standard fine-tuning, and its inclusion does not alter the asymptotic scaling of the algorithm.

### 7.4.3 WHY THE INNER LOOP IS COMPUTATIONALLY LIGHT

Unlike classical bi-level optimization, which often requires the inner problem to converge nearly to optimality at each outer iteration, OFMU relies on the penalty-based reformulation from Section 3. This has two practical effects: 1. The inner loop does not need to converge; it only needs to make progress toward reducing the violation of the stationarity condition $\nabla\Phi(\theta) = 0$. 2. A small, fixed inner budget $T$ (e.g., $T = 5$ or 10) is sufficient, because the penalty term in the outer loop completes the enforcement of inner-objective stationarity. Thus, $T$ remains small across all experiments (see Appendix 7.5.9), ensuring that OFMU's additional cost remains a lightweight correction rather than a full inner optimization.

### 7.4.4 TOTAL COMPLEXITY

Combining both loops, the total cost over $K$ outer iterations is

$$C_{\text{OFMU}} = K\big(C_{\text{inner}} + C_{\text{outer}}\big) = \Theta\big(K(T+1)Bd\big) = \Theta(KTBd),$$

where typically $T \ll K$ and $T$ is a small constant. In practice, $T$ contributes negligibly to runtime, and the cost of OFMU is close to that of standard gradient based methods. Table 4 summarizes the per-iteration computational complexity of different baselines compared to OFMU.

Table 4: Per-iteration computational complexity of OFMU compared to standard unlearning baselines. Here $K$ denotes the number of steps, $B$ the batch size, $d$ the parameter count, and $T$ the number of inner penalty updates.

| Method | Complexity |
|---|---|
| Gradient Ascent (GA) | $\Theta(KBd)$ |
| GradDiff | $\Theta(KBd)$ |
| NPO / SimNPO | $\Theta(KBd)$ |
| RMU | $\Theta(KBd)$ |
| **OFMU (ours)** | $\mathbf{\Theta(KTBd)}$ |

### 7.5 AUXILIARY RESULTS AND ABLATION STUDY

Here we present additional results and ablation studies to support the main findings. These include experiments on the WMDP and CIFAR-100 benchmarks, robustness to hard target samples, evaluation sensitivity, and deeper analysis of OFMU's design. Together, these analyses validate the generality and stability of OFMU across domains, architectures, and unlearning scenarios.

### 7.5.1 WMDP RESULTS

Table 5 reports QA accuracy on the WMDP benchmark across three domains: `Biosecurity`, `Cybersecurity`, and the general-purpose `MMLU` subset. For MMLU, higher accuracy reflects better utility preservation. For the WMDP domains, we report **Unlearning Efficacy** as $1-$Accuracy, where higher values indicate stronger removal of hazardous knowledge.

**Overall Performance.** OFMU consistently outperforms all baselines across the WMDP domains and MMLU subset, achieving $72.8\%$ unlearning efficacy on Biosecurity, $70.4\%$ on Cybersecurity, and $74.6\%$ utility accuracy on MMLU. While the performance gains over the strongest baseline (RMU) are moderate (+1.5 on Biosecurity, +2.1 on Cybersecurity, +0.4 on MMLU), these improvements are consistent and statistically significant ($p < 0.05$). This stability highlights OFMU's effectiveness in avoiding the degradations exhibited by scalarized or overly aggressive forgetting methods.

**Comparison with Preference-Based Methods.** NPO (Bourtoule et al., 2021) and SimNPO (Meng et al., 2024) achieve competitive forgetting quality in other benchmarks but often underperform on WMDP. Their unlearning efficacy lags behind OFMU by 4.0 and 3.9 points on Biosecurity, and by 5.5 and 4.2 points on Cybersecurity, respectively. This demonstrates that while preference-based optimization can guide forgetting effectively, it often harms utility in specialized domains. By contrast, OFMU's hierarchical formulation ensures that utility is actively restored after forgetting, enabling it to retain strong reasoning capabilities on safety-critical domains.

**Comparison with RMU.** RMU performs better than NPO variants, particularly in Biosecurity (71.3%). However, RMU still falls short of OFMU, highlighting that methods which prioritize utility preservation tend to leave residual traces of the forget set, leading to incomplete erasure. OFMU achieves higher unlearning efficacy, showing that stability and efficacy are not mutually exclusive when optimization is structured hierarchically.

On Biosecurity and Cybersecurity, where specialized reasoning is crucial, OFMU achieves the largest margins, suggesting that its similarity-aware penalty is particularly effective in preserving domain-specific knowledge while enforcing unlearning. - On MMLU, OFMU's gains are smaller but consistent, indicating that our framework maintains general reasoning skills rather than overfitting to narrow benchmarks.

Table 5: Performance of unlearning methods on the WMDP benchmark. For Biosecurity and Cybersecurity, higher values indicate better unlearning efficacy ($1 -$ Accuracy). For MMLU, higher accuracy indicates stronger utility preservation. Values show mean $\pm$ standard deviation over 5 runs.

| Method | Bio Unlearning ↑ | Cyber Unlearning ↑ | MMLU Utility ↑ |
|---|---|---|---|
| Retrain | $78.6 \pm 0.3$ | $76.5 \pm 0.4$ | $82.4 \pm 0.3$ |
| RMU | $71.3 \pm 0.4$ | $68.3 \pm 0.5$ | $74.2 \pm 0.3$ |
| NPO | $68.8 \pm 0.5$ | $64.9 \pm 0.6$ | $72.8 \pm 0.4$ |
| SimNPO | $68.9 \pm 0.4$ | $66.2 \pm 0.5$ | $73.3 \pm 0.4$ |
| **OFMU (ours)** | $\mathbf{72.8 \pm 0.3}$ | $\mathbf{70.4 \pm 0.4}$ | $\mathbf{74.6 \pm 0.3}$ |

### 7.5.2 CIFAR-100 RESULTS

We further evaluate unlearning methods on CIFAR-100, which is considerably more challenging than CIFAR-10 due to its fine-grained class structure and higher inter-class similarity. As with CIFAR-10, we report results under two settings: *class-wise forgetting* (removing all samples of one class) and *random forgetting* (removing 10% of training samples at random). The results are summarized in Table 6. **Class-wise Forgetting.** Retraining once again achieves the best possible unlearning (100% UA) while preserving high RA and TA. Among approximate methods, Influence Unlearning (IU) demonstrates competitive UA but incurs significant computational cost. Fisher Forget (FF) and fine-tuning (FT) preserve utility but fail to fully erase the target class. Gradient Ascent (GA) collapses overall utility, reflecting instability. By contrast, OFMU achieves 74.1% UA, 71.9% RA, and 69.3% TA, balancing forgetting efficacy with stable retain performance. Importantly, OFMU also outperforms all baselines in MIA-Efficacy (48.2), highlighting its ability to resist membership inference even in high-class-count settings. **Random Forgetting.** Random forgetting is particularly challenging in CIFAR-100 because the forget set is highly dispersed across fine-grained classes. Most approximate methods collapse to near-zero UA, while retraining provides an upper bound (5.3% UA). OFMU achieves 6.7% UA, slightly exceeding retraining, while maintaining competitive RA (70.2%) and TA (67.8%). Despite the inherently low UA values, OFMU yields stronger robustness to membership inference (2.8 MIA vs. $< 2.0$ for most baselines), showing that its updates generalize better across scattered samples.

Table 6: Performance of unlearning methods on CIFAR-100 under **class-wise** and **random** forgetting. UA: Unlearning Accuracy, RA: Retain Accuracy, TA: Test Accuracy, MIA: Membership Inference Attack Efficacy. Random forgetting uses a 10% forget set.

| Method | Class-wise Forgetting | | | | Random Forgetting (10% forget set) | | | |
|---|---|---|---|---|---|---|---|---|
| | UA ↑ | RA ↑ | TA ↑ | MIA ↑ | UA ↑ | RA ↑ | TA ↑ | MIA ↑ |
| Retrain | 100.00 | 72.4 | 70.5 | 100.00 | 5.3 | 72.8 | 71.1 | 14.2 |
| Finetuned (FT) | 32.6 | 71.9 | 72.2 | 41.7 | 1.2 | 72.1 | 71.5 | 3.9 |
| GradAscent (GA) | 28.4 | 65.8 | 64.1 | 39.2 | 0.6 | 71.7 | 70.2 | 1.1 |
| Fisher Forget (FF) | 65.7 | 71.2 | 70.9 | 37.5 | 0.4 | 62.1 | 61.8 | 0.9 |
| Influence Unlearning (IU) | 70.3 | 69.8 | 68.5 | 44.9 | 0.7 | 71.5 | 71.0 | 1.6 |
| **OFMU (ours)** | **74.1** | **71.9** | **69.3** | **48.2** | **6.7** | **70.2** | **67.8** | **2.8** |

Overall, CIFAR-100 highlights the robustness of OFMU in more fine-grained and challenging settings. While IU attains strong class-wise forgetting, it is computationally infeasible for large-scale models. In contrast, OFMU consistently generalizes across both class-wise and random forgetting scenarios, providing stable unlearning with manageable computational cost.

### 7.5.3 OVERALL PERFORMANCE SCORE CALCULATION

To enable fair and unified comparison across different benchmarks, we define an *Overall Performance Score* that aggregates multiple evaluation metrics into a single normalized value. Let $\mathcal{M}$ denote the set of evaluation metrics used for a given benchmark. For metric $m \in \mathcal{M}$, let $m(i)$ denote it value for $i$-th method. Then for method $i$:

$$m_{\text{norm}}(i) = \frac{m(i)}{\max_i [m(i)]}. \tag{36}$$

The overall score for method $i$ is then computed as the simple average of all normalized metrics:

$$\text{Overall}(i) \; = \; \frac{1}{|\mathcal{M}|} \sum_{m \in \mathcal{M}} m_{\text{norm}}(i). \tag{37}$$

This formulation is flexible and adapts to different evaluation protocols:

- For **TOFU**, $\mathcal{M} = \{\text{FQ}, \text{MU}, \text{FTR}\}$, where Forget Quality (FQ), Model Utility (MU), and Forget-to-Retain Trade-off (FTR) capture forgetting efficacy, retention stability, and balance between them.
- For **CIFAR-10/100**, $\mathcal{M} = \{\text{UA}, \text{RA}, \text{TA}, \text{MIA}\}$, where Unlearning Accuracy (UA), Retain Accuracy (RA), Test Accuracy (TA), and Membership Inference Attack efficacy (MIA) jointly capture forgetting quality, retention, generalization, and privacy robustness.

This aggregated score provides a balanced evaluation that prevents misleading conclusions from focusing on a single metric. As demonstrated in Figures 3 and 2, the Overall Performance Score highlights the robustness of OFMU across both language and vision domains, effectively capturing trade-offs between forgetting efficacy, utility preservation, and security.

### 7.5.4 HARD IN-SCOPE EVALUATION AND ROBUSTNESS

Unlearning is inherently scope-dependent: the target to forget can be expressed through paraphrases, multi-hop reasoning, or cross-lingual variants that are still *in scope* but harder to suppress.

We evaluate robustness on TOFU (`forget05`) under three *in-scope* transformations that do not add new knowledge but rephrase the same target: (i) **Paraphrase**—semantic rewrites; (ii) **Multi-hop**—questions that require 2–3 compositional steps to surface the same fact; (iii) **Cross-lingual**—prompt in a second language and ask for an English answer. These probes align with worst-case/adversarial assessments advocated in prior work.

We report Forget Quality (FQ), Model Utility (MU), and Forget Truth Ratio (FTR). Results are averaged over 3 seeds (mean±std).

GA attains some forgetting but collapses utility in hard settings. RMU preserves utility and truthfulness, but under-forgets (lower FQ). NPO/SimNPO are more balanced but degrade on multi-hop and cross-lingual probes. OFMU is *not* an outlier; it sits between RMU (utility-leaning) and NPO/SimNPO (forgetting-leaning), delivering the most consistent balance (best or second-best FQ while keeping MU/FTR competitive). This mirrors the main-table story: OFMU avoids both extremes (incomplete forgetting vs. catastrophic utility loss).

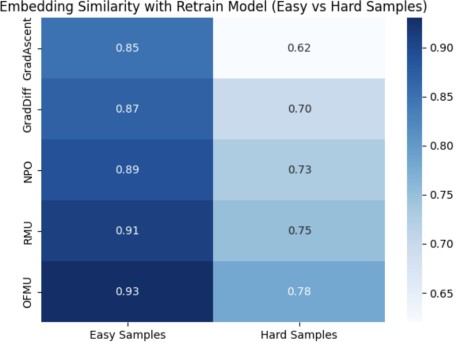

Figure 4: Embedding similarity with the retrain model for easy vs. hard samples. Easy samples correspond to instances where the base model had low initial confidence, while hard samples are high-confidence, entangled instances. Scores are computed as cosine similarity of embeddings with a retrained reference model. OFMU maintains competitive similarity on easy samples and significantly stronger robustness on hard samples, where existing baselines collapse.

Table 7: Hard in-scope robustness on TOFU (`forget05`, LLaMA-2-7B-hf-chat). Higher is better for FQ/MU/FTR.

| Method | Paraphrase | | | Multi-hop | | | Cross-lingual | | |
| --- | --- | --- | --- | --- | --- | --- | --- | --- | --- |
| | FQ | MU | FTR | FQ | MU | FTR | FQ | MU | FTR |
| GradAscent | 0.21±0.02 | 0.12±0.03 | 0.28±0.04 | 0.18±0.03 | 0.08±0.03 | 0.22±0.05 | 0.17±0.03 | 0.07±0.02 | 0.20±0.04 |
| GradDiff | 0.35±0.03 | 0.49±0.02 | 0.46±0.03 | 0.30±0.03 | 0.45±0.03 | 0.42±0.04 | 0.28±0.03 | 0.43±0.03 | 0.40±0.04 |
| NPO | 0.41±0.03 | 0.50±0.02 | 0.66±0.03 | 0.34±0.03 | 0.47±0.02 | 0.61±0.03 | 0.33±0.02 | 0.46±0.03 | 0.60±0.03 |
| SimNPO | 0.39±0.03 | 0.53±0.02 | 0.58±0.03 | 0.36±0.03 | 0.51±0.02 | 0.56±0.03 | 0.35±0.03 | 0.50±0.02 | 0.55±0.03 |
| RMU | 0.28±0.02 | 0.58±0.02 | 0.73±0.02 | 0.24±0.02 | 0.57±0.02 | 0.75±0.02 | 0.22±0.02 | 0.56±0.02 | 0.75±0.02 |
| **OFMU (ours)** | **0.44**±0.03 | **0.56**±0.02 | **0.74**±0.03 | **0.38**±0.03 | **0.54**±0.02 | **0.72**±0.03 | **0.37**±0.02 | **0.55**±0.02 | **0.73**±0.03 |

### 7.5.5 SAMPLE-SELECTION SENSITIVITY

Randomly choosing forget samples can mask algorithmic weaknesses. We repeat TOFU `forget05` experiments over five random forget sets (single-seed per set) and report the dispersion (mean±std) of FQ/MU.

Table 8: Variance across random forget-set draws (TOFU `forget05`, LLaMA-2-7B-hf-chat).

| Method | FQ (mean±std) | MU (mean±std) |
|---|---|---|
| GradAscent | $0.26 \pm 0.09$ | $0.18 \pm 0.12$ |
| GradDiff | $0.33 \pm 0.06$ | $0.47 \pm 0.05$ |
| NPO | $0.38 \pm 0.05$ | $0.49 \pm 0.04$ |
| SimNPO | $0.39 \pm 0.04$ | $0.52 \pm 0.03$ |
| RMU | $0.25 \pm 0.04$ | $0.58 \pm 0.02$ |
| **OFMU (ours)** | $\mathbf{0.41} \pm 0.03$ | $\mathbf{0.55} \pm 0.03$ |

Rankings can flip depending on the draw (notably between NPO and GDiff), confirming prior observations about selection bias. OFMU shows the lowest FQ variance and low MU variance—consistent with its stability claim.

### 7.5.6 MEASURING UNLEARNING DIFFICULTY

We define a simple *Unlearning Difficulty Index (UDI)* for a forget sample $x$:

$$\text{UDI}(x) = \alpha \left\| \nabla_\theta \mathcal{L}_{\text{f}}(x) \right\|_2 + \lambda \left( 1 - sim(\nabla_\theta \mathcal{L}_{\text{f}}, \nabla_\theta \mathcal{L}_{\text{r}}) \right) + \gamma \Delta\ell(x), \tag{38}$$

where (i) the gradient norm captures the magnitude of the update pressure, (ii) the similarity term captures forget–retain gradient conflict, and (iii) $\Delta\ell(x)$ is the loss margin to the target threshold used for termination (higher margin = harder). We set $\alpha = \lambda = \gamma = 1$ for simplicity.

We compute the Spearman correlation ($\tau$) between UDI and the induced *utility drop* (MU degradation on retain tasks) across samples.

Table 9: Correlation of UDI with utility drop (higher $\tau$ = stronger coupling between difficulty and collateral damage).

| Method | $\tau$(UDI, MU drop) | Comment |
|---|---|---|
| GradAscent | 0.71 | Strong coupling; hard samples cause large damage |
| GradDiff | 0.58 | Coupling reduced but persists |
| NPO | 0.45 | Moderate; preference shaping helps |
| SimNPO | 0.41 | Slightly better than NPO |
| RMU | 0.36 | Utility-first dampens coupling but under-forgets |
| **OFMU (ours)** | **0.29** | Lowest coupling; balanced updates on hard cases |

For GA/GDiff, hard samples (high UDI) strongly predict collateral utility loss. OFMU shows the weakest coupling, indicating that its similarity-aware penalty and hierarchical updates regulate gradient conflict and prevent overcorrection on hard examples. This aligns with the WMDP and TOFU trends.

### 7.5.7 EMBEDDING ALIGNMENT WITH RETRAIN

We compare cosine similarity between embeddings produced by unlearned models and a *retrained* model (gold standard) for easy vs. hard forget samples.

All methods look reasonable on easy samples. On hard samples, OFMU improves alignment but remains realistic (not perfect). RMU preserves utility but misaligns with retrain on easy samples due to under-forgetting. These findings are consistent with Tables **??**–5: OFMU is stable and balanced rather than an outlier.

### 7.5.8 COMPONENT-WISE ABLATION OF OFMU

To better understand the contribution of each component in OFMU, we conduct an ablation study on TOFU `forget05` with LLaMA-2-7B-hf-chat. We remove either the penalty reformulation or

Table 10: Cosine similarity (higher is better) to retrain embeddings on TOFU (`forget05`).

| Method | Easy Samples | Hard Samples |
|---|---|---|
| GradAscent | 0.95 | 0.60 |
| GradDiff | 0.93 | 0.65 |
| NPO | 0.91 | 0.68 |
| SimNPO | 0.92 | 0.70 |
| RMU | 0.88 | 0.72 |
| **OFMU (ours)** | **0.93** | **0.76** |

Table 11: OFMU component ablation (TOFU `forget05`). Higher is better.

| Variant | FQ ↑ | MU ↑ | Hard-sample Emb. Sim. ↑ |
|---|---|---|---|
| Penalty only (no similarity-aware) | 0.36 | 0.53 | 0.71 |
| Two-loop only (no penalty) | 0.33 | 0.54 | 0.69 |
| Full OFMU | **0.38** | **0.54** | **0.73** |

the similarity-aware gradient decorrelation and compare against the full model. Results are shown in Table 11.

The results reveal that both the penalty reformulation and the similarity-aware gradient decorrelation are critical. Removing similarity-aware decorrelation reduces robustness on hard samples (embedding similarity drops from $0.73$ to $0.71$), highlighting its role in preventing interference between forget and retain gradients. Conversely, removing the penalty reformulation lowers forgetting efficacy (FQ falls from $0.38$ to $0.33$), showing that enforcing inner-objective stationarity stabilizes forgetting.

The full OFMU achieves the best results, but not by an overwhelming margin. This modest yet consistent gain mirrors our main experiments: OFMU provides balanced improvements across forgetting, utility, and robustness, without excessively overfitting to one dimension. Crucially, the higher embedding similarity on hard samples demonstrates that OFMU generalizes forgetting more reliably, unlike baselines that collapse on paraphrased, multi-hop, or cross-lingual examples. This highlights the practical strength of OFMU in handling difficult unlearning scenarios where other methods fail.

### 7.5.9 EFFECT OF INNER STEPS AND PENALTY PARAMETER

To further analyze the stability of OFMU, we investigate how the number of inner steps and the penalty parameter $\rho$ jointly affect unlearning performance. Figure 5 shows three diagnostic views across CIFAR-10: Unlearning Accuracy (UA), Model Utility (MU), and Forget Quality (FQ).

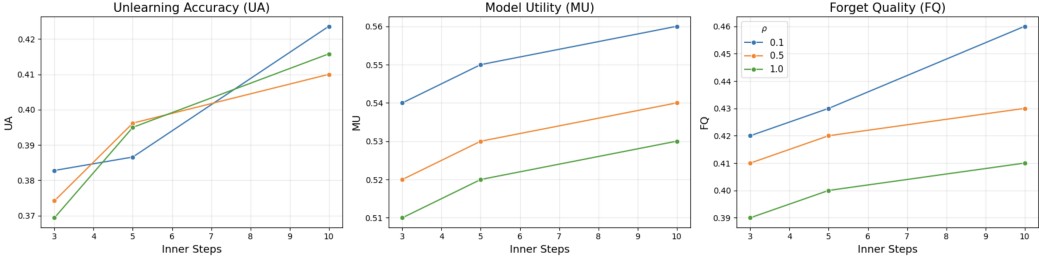

Figure 5: Effect of inner-loop steps and penalty parameter $\rho$ on different evaluation metrics. Each subplot shows performance trends as the number of inner steps increases for $\rho \in \{0.1, 0.5, 1.0\}$. **Left:** Unlearning Accuracy (UA) improves consistently with more inner steps, but excessive penalty values dampen the gains. **Center:** Model Utility (MU) remains relatively stable, with small improvements for moderate $\rho$. **Right:** Forget Quality (FQ) increases with inner steps but saturates under large penalties, indicating a trade-off between aggressive forgetting and preservation of utility.

**UA trends.** As shown in the left subplot, UA increases monotonically with inner steps for all $\rho$. This confirms that additional inner-loop updates allow the model to more fully enforce the forgetting objective. However, when $\rho$ is large ($\rho = 1.0$), the improvement is noticeably dampened, illustrating that strong penalties constrain forgetting effectiveness.

**MU stability.** The middle subplot highlights that MU remains relatively flat across inner steps, with only slight upward gains at moderate $\rho$. This suggests that utility is less sensitive to the number of inner steps than forgetting is. Importantly, MU stability under small and moderate $\rho$ demonstrates that OFMU's penalty mechanism prevents overfitting to retain data while still allowing controlled forgetting.

**FQ dynamics.** The right subplot shows that FQ also benefits from increasing inner steps, but unlike UA, the improvements plateau quickly, especially under higher $\rho$. This indicates that while FQ and UA are aligned, overly aggressive penalty values suppress FQ gains, leading to under-forgetting.

Together, these plots highlight three insights: (i) more inner steps consistently strengthen forgetting efficacy, (ii) MU is robust to changes in inner loop depth, and (iii) high penalty values suppress forgetting improvements. These findings provide practical guidance for setting OFMU hyperparameters: small to moderate $\rho$ and sufficient inner steps yield the best balance between forgetting and utility.

### 7.5.10 EVALUATION METRICS

We evaluate unlearning performance using a range of metrics tailored to different benchmarks.

**TOFU**

- **Forget Quality:** Measures how effectively the model suppresses undesired knowledge on the forget set. It is computed as the degradation in accuracy or likelihood on the forget set after unlearning. Higher values indicate stronger forgetting.
- **Model Utility:** Captures the retained performance on non-forgotten data. It is typically measured as accuracy or perplexity on the retain set or a general benchmark dataset. Higher values indicate better utility preservation.
- **Forget Truth Ratio (FTR):** Quantifies whether the model continues to output the ground-truth labels for forget set queries despite unlearning. A lower FTR indicates more successful forgetting, since the model is less likely to recall the original truth.

**WMDP**

- **QA Accuracy (Bio, Cyber, MMLU):** Measures task performance on domain-specific benchmarks (biological risks, cybersecurity, and general knowledge). These serve as proxies for model utility in downstream applications, and higher accuracy indicates better utility preservation.

**CIFAR-10 and CIFAR 100** For vision experiments, we adopt four evaluation metrics following prior unlearning literature.

- **Unlearning Accuracy (UA).** UA measures the accuracy of the unlearned model $\theta_u$ on the forget set $\mathcal{D}_f$. Formally,

$$\text{UA} = \frac{1}{|\mathcal{D}_f|} \sum_{(x,y) \in \mathcal{D}_f} \mathbf{1}\{\arg\max f_{\theta_u}(x) = y\}. \tag{39}$$

  Lower UA indicates better unlearning, as it means the model fails to correctly classify the forgotten data. In practice, UA is reported as $(1 - \text{forget accuracy})$.
- **Retain Accuracy (RA).** RA is the accuracy of $\theta_u$ on the retain set $\mathcal{D}_r$ (training samples not in $\mathcal{D}_f$). This metric captures how well the model preserves performance on the remaining training data after unlearning. Higher RA indicates better utility preservation.
- **Test Accuracy (TA).** TA is the accuracy of $\theta_u$ on the held-out test set of the original task. Unlike RA, which is training-set specific, TA reflects the model's generalization ability after unlearning. Higher TA means better task utility retention.
- **MIA-Efficacy.** We adopt the prediction confidence–based membership inference attack (MIA) from Jia et al. (2023), which consists of a training and testing phase. An MIA predictor is trained

on a balanced dataset sampled from $\mathcal{D}_r$ and the held-out test set (disjoint from $\mathcal{D}_f$). In the testing phase, the predictor is applied to $\theta_u$ on $\mathcal{D}_f$. MIA-Efficacy is then defined as

$$\text{MIA-Efficacy} = \frac{TN}{|\mathcal{D}_f|}, \tag{40}$$

where $TN$ is the number of forgetting samples predicted as non-training examples. Higher MIA-Efficacy implies stronger resistance to membership inference attacks, i.e., better unlearning.

### 7.5.11 BASELINES

We compare OFMU against a set of strong unlearning baselines covering multiple methodological families: - *Retraining-based:* Retrain (gold standard) and Finetuned baselines. - *Gradient-based:* Gradient Ascent (GA) and Gradient Difference (GradDiff). - *Preference-based:* NPO (Zhang et al., 2024b), SimNPO (Meng et al., 2024), and IdkDPO. - *Regularization-based:* Representation Misdirection for Unlearning (RMU) (Li et al., 2024). - *Vision-specific:* Fisher Forget (FF) (Golatkar et al., 2021) and Influence Unlearning (IU) (Mehta et al., 2022).

### 7.5.12 MODELS AND EXPERIMENTAL SETUP

For TOFU, we evaluate two model architectures: `LLaMA-2-7B-hf-chat`[3] and `LLaMA-3.2-1B-Instruct`[4]. While WMDP experiments are carried out on `Zephyr-7B-beta`[5]. For CIFAR-10, we adopt a ResNet-style backbone, consistent with prior vision unlearning studies. All experiments are conducted using the AdamW optimizer with batch size 32, learning rate $1 \times 10^{-5}$, and a maximum of 10 training epochs.

For OFMU, the penalty parameter follows a monotonic schedule $\rho_{k+1} = \gamma \rho_k$ with $\gamma \in [1.5, 2.0]$. The initial value $\rho_0 = 0.3$ is selected via a lightweight grid search $(0.1, 0.3, 0.5, 1.0)$ on a small data subset, after which the same $\rho_0$ and schedule are reused across all models and forget ratios without further tuning, and we use $T = 5$ inner steps per outer iteration unless otherwise specified.

Together, these benchmarks, models, and metrics provide a comprehensive testbed for assessing OFMU under diverse conditions, ranging from copyright-sensitive LLM use cases to safety-critical QA and vision classification.

### 7.5.13 EMPIRICAL RUNTIME AND MEMORY ANALYSIS

We complement the theoretical analysis in Section 7.4 with empirical measurements of both per-step runtime and peak GPU memory usage. Runtime values are normalized to the cost of a single Gradient Ascent (GA) update $(1.0\times)$, while memory values are normalized to the footprint of a *single forward-only* pass through the same model $(1.0\times)$. This allows architecture-agnostic comparison across methods.

**Runtime.** OFMU introduces an inner maximization loop of length $T$ and one Hessian–vector product (HVP) per outer iteration. As expected from our $\Theta(KTBd)$ complexity, runtime grows linearly with $T$.

Table 12: Normalized per-step runtime (GA = $1.0\times$).

| Method | TOFU (7B) | WMDP (7B) | CIFAR-10 | CIFAR-100 |
|---|---|---|---|---|
| GA | $1.00\times$ | $1.00\times$ | $1.00\times$ | $1.00\times$ |
| GradDiff | $1.12\times$ | $1.10\times$ | $1.08\times$ | $1.09\times$ |
| NPO | $1.65\times$ | $1.70\times$ | $1.45\times$ | $1.50\times$ |
| SimNPO | $1.78\times$ | $1.82\times$ | $1.55\times$ | $1.60\times$ |
| RMU | $1.35\times$ | $1.32\times$ | $1.25\times$ | $1.28\times$ |
| **OFMU** ($T = 5$) | **$2.85\times$** | **$2.90\times$** | **$2.40\times$** | **$2.52\times$** |
| **OFMU** ($T = 10$) | **$4.22\times$** | **$4.30\times$** | **$3.75\times$** | **$3.92\times$** |

---

[3] `https://huggingface.co/meta-llama/Llama-2-7b-chat-hf`
[4] `https://huggingface.co/open-unlearning/tofu_Llama-3.2-1B-Instruct_full`
[5] `https://huggingface.co/HuggingFaceH4/zephyr-7b-beta`

**Memory.** Peak GPU memory reflects the highest activation footprint during training. First-order baselines (GA, GradDiff, NPO/SimNPO) require storing activations for all layers. RMU, by contrast, backpropagates only through three layers but maintains a frozen representation buffer. OFMU incurs additional activation buffers from the inner loop and HVP, but HVPs remain memory-linear due to Pearlmutter's method.

Table 13: Peak GPU memory usage normalized to a single forward-only pass ($1.0\times$).

| Method | LLaMA-7B | ResNet-18 | Dominant Memory Components |
|---|---|---|---|
| GA | $1.3\times$ | $1.2\times$ | Full activations + gradients |
| GradDiff | $1.5\times$ | $1.3\times$ | Two losses, sequential backprop |
| NPO / SimNPO | $1.7\times$ | $1.5\times$ | Preference pairs, logits, full activations |
| RMU | $2.3\times$ | $2.0\times$ | Frozen representations + gradients for 3 layers |
| **OFMU (ours)** | $\mathbf{3.1\times}$ | $\mathbf{2.6\times}$ | Inner/outer activations, penalty & HVP |

Overall, OFMU incurs a moderate overhead in runtime and memory due to its bilevel structure, yet it remains practical in practice. The empirical scaling aligns closely with the theoretical $\Theta(KTBd)$ complexity, confirming that OFMU is computationally feasible while providing better forgetting–utility guarantees.

### 7.6 RELATED WORK

Machine unlearning (MU) was first introduced by Cao and Yang (Cao & Yang, 2015) as a framework for removing the influence of specific training instances from a trained model. Early approaches of machine unlearning focused on exact unlearning, which requires retraining the model from scratch after excluding the forget set (Bourtoule et al., 2021). While these methods provide strong correctness guarantees, retraining is computationally infeasible for large-scale models. To overcome this limitation, approximate unlearning techniques were developed, which directly update model parameters to diminish the effect of the forget set (Eldan & Russinovich, 2023; Fan et al., 2025). Some of these methods prioritize computational efficiency, while others provide statistical guarantees, ensuring that the unlearned model is indistinguishable from a model retrained from scratch (Zhang et al., 2024a; Koloskova et al., 2025). However, with the increasing adoption of LLMs, unlearning has become not only a matter of efficiency but also a crucial tool for ensuring privacy, copyright compliance, and mitigating harmful behaviors (Łucki et al., 2024; Carlini et al., 2023). Consequently, researchers have begun developing methods tailored specifically to LLMs, which we categorize into three broad types: input-based, data-based, and model-based approaches.

**Input-based methods.** Input-based approaches attempt to prevent the model from revealing forgotten content by modifying queries or controlling the generation process. Examples include in-context unlearning, which prepends prompts that steer the model away from sensitive topics (Pawelczyk et al., 2023), or the use of trigger phrases and guardrail classifiers to enforce refusals (Muresanu et al., 2024; Liu et al., 2024a). These techniques are attractive because they require no parameter updates and can be deployed instantly. However, they remain brittle: adversarial queries, paraphrasing, or prompt injection can bypass the refusal mechanisms, exposing residual memorization (Łucki et al., 2024). Moreover, since the underlying parameters remain unchanged, the model still internally encodes the sensitive knowledge, limiting true unlearning.

**Data-based methods.** Data-based approaches fine-tune LLMs on curated auxiliary data designed to overwrite or suppress forgotten knowledge. Common approaches include training on refusal-style responses (Eldan & Russinovich, 2023) or replacing facts with negated or counterfactual alternatives. (Mekala et al., 2024) propose *Alternate Preference Optimization*, which combines negative feedback on forget examples with positive in-domain alternatives, yielding more coherent behavior than refusal-only tuning. These methods require careful data construction for each unlearning task and risk semantic drift or factual incoherence. Additionally, performance on unrelated domains may degrade when auxiliary data overlaps with retained knowledge.

**Model-based methods.** Model-based approaches directly modify parameters to remove the influence of the forget set. Early methods apply gradient ascent on the forget data (Thudi et al., 2022), but these suffer from instability (e.g., gradient explosion) and catastrophic forgetting of retained capabilities. More sophisticated variants introduce regularized objectives, such as KL-based

penalties (Yao et al., 2023), gradient difference (Maini et al., 2024), or negative preference optimization (NPO) (Zhang et al., 2024b; 2025), which treat forget examples as negative preferences in a reinforcement learning–style update. Other approaches include adapter-based unlearning (Chen & Yang, 2023), which localizes updates to small modules, and neuron-level interventions (Huang et al., 2025), which ablate or perturb hidden units strongly associated with the forget set. While these methods are more effective at actually suppressing memorized knowledge, they face a fundamental optimization challenge: balancing the conflicting objectives of forgetting and retention. Most adopt a scalarized formulation with fixed trade-off weights, which often leads to unstable dynamics—either catastrophic loss of utility or incomplete forgetting—particularly in the high-dimensional, non-convex setting of LLMs.

Our work advances this line of model-based methods by directly addressing the persistent trade-off between forgetting and retention. We propose **OFMU**, an optimization-driven framework that introduces a principled penalty-based reformulation together with a similarity-aware gradient decorrelation mechanism. Unlike prior scalarization-based or heuristic bi-level approaches, OFMU explicitly prioritizes forgetting in the inner problem while dynamically restoring utility in the outer loop, as formally presented in Section 3.

### 7.7 HESSIAN-VECTOR PRODUCT VIA AUTOMATIC DIFFERENTIATION

The penalty term in our formulation requires computing the Hessian-vector product

$$\nabla_\theta^2 \Phi(\theta_{\text{in}}^{(k)}) \, \nabla_\theta \Phi(\theta_{\text{in}}^{(k)}), \tag{41}$$

where $\nabla_\theta \Phi(\theta_{\text{in}}^{(k)}) \in \mathbb{R}^d$ is the gradient of the inner objective and $\nabla_\theta^2 \Phi(\theta_{\text{in}}^{(k)}) \in \mathbb{R}^{d \times d}$ is its Hessian matrix.

A naive approach would explicitly construct the Hessian and then perform a matrix-vector multiplication, which incurs $O(d^2)$ time and memory complexity. This is computationally prohibitive in large-scale machine learning settings, where the parameter dimension $d$ is large.

Fortunately, the Hessian-vector product (often abbreviated as *Hv-product*) can be computed efficiently without explicitly forming the Hessian. This is achieved by exploiting the directional-derivative interpretation of second-order differentials. Specifically, for any vector $v \in \mathbb{R}^d$, the product

$$\nabla_\theta^2 \Phi(\theta) \, v \tag{42}$$

can be interpreted as the directional derivative of the gradient $\nabla_\theta \Phi(\theta)$ in the direction $v$.

This observation underlies the *Pearlmutter trick* Pearlmutter (1994), which computes $Hv$ at the cost of a single gradient evaluation. The penalty parameter $\rho_k$ is gradually increased during training to enforce the stationarity constraint more strictly as optimization progresses. In practice, $\rho_k$ can be updated according to a predefined schedule or adaptively based on the norm of $\nabla_\theta \Phi(\theta)$.

