# OpenReview forum: "OFMU: OPTIMIZATION-DRIVEN FRAMEWORK FOR MACHINE UNLEARNING"
_ICLR.cc/2026/Conference — ICLR 2026 Poster_

### Official Review · Reviewer_9aX8 · 2025-10-26

**Soundness:** 3
**Presentation:** 3
**Contribution:** 3
**Rating:** 6
**Confidence:** 3

**Summary:**

This paper proposes OFMU, a bi-level optimization framework for machine unlearning in large language models. The key idea is to formulate unlearning as a hierarchical optimization problem where the inner loop maximizes forgetting (with a similarity-aware gradient decorrelation penalty) and the outer loop minimizes retain loss. The authors develop a penalty-based reformulation to avoid expensive nested optimization, provide convergence analysis for both convex and non-convex settings, and validate their approach on TOFU, WMDP (language) and CIFAR-10/100 (vision) benchmarks.

The method shows consistent but modest improvements over existing baselines like Gradient Ascent, NPO, and RMU, particularly in maintaining stability on hard-to-forget samples and achieving better balance between forgetting quality and model utility.

**Strengths:**

S1: Well-motivated framework

The hierarchical formulation clearly captures the asymmetry between forgetting (must succeed) and retention (restore afterward). The similarity-aware penalty that decorrelates forget and retain gradients addresses a real problem in existing scalarization approaches.

S2: Solid theoretical grounding

Lemmas 1-3 provide rigorous convergence analysis for both convex and non-convex settings. The penalty reformulation (Equation 7) is theoretically justified and avoids the computational burden of fully solving the inner problem at each iteration.

S3: Consistent empirical performance

OFMU shows stable improvements across multiple benchmarks and scenarios. I particularly like Figure 1's analysis showing reduced coupling between sample difficulty and utility loss (though this might be from appendix content—focusing on main text results). The method avoids the catastrophic utility collapse of Gradient Ascent while achieving better forgetting than utility-focused methods like RMU.

S4: Comprehensive evaluation

The paper evaluates across both language (TOFU, WMDP) and vision (CIFAR-10) tasks with multiple metrics capturing different aspects of unlearning quality.

**Weaknesses:**

W1: Modest and potentially insignificant improvements

Looking at Table 1, the gains are small and inconsistent. For forget05 on LLaMA-2, OFMU gets FQ=0.13 vs NPO's 0.09, but NPO actually beats OFMU on forget10 (0.42 vs 0.41). More critically, the paper provides no error bars or significance tests in main results tables. Without confidence intervals, it's impossible to judge whether these differences are meaningful or just noise.

W2: Computational cost completely unaddressed

The method requires Hessian-vector products (Equation 9) at each outer iteration, plus T inner gradient ascent steps. This is clearly more expensive than single-loop baselines, yet the paper provides zero analysis of wall-clock time, memory usage, or FLOPs. For a method claiming "scalability" (abstract), this is a major omission that undermines practical applicability.

W3: Limited scale and missing ablations

The largest model tested is LLaMA-2-7B. Given that the method involves second-order computations, scalability to 70B+ models is questionable but unexplored. Additionally, key design choices lack justification—why T inner steps specifically? How sensitive is performance to β and ρ schedules? These are mentioned but not systematically studied in the main paper.

**Questions:**

Q1: Statistical significance

Can you provide confidence intervals or statistical tests for the main results in Tables 1-3? Given the small effect sizes, this is essential to determine if improvements are real or within noise margins.

Q2: Computational overhead

What is the actual training time and memory consumption of OFMU compared to baselines like GA and NPO? How many additional gradient computations does the Hessian-vector product require per iteration?

Q3: Scalability validation

Have you tested on models larger than 7B parameters? If not, what are the expected computational bottlenecks that would prevent scaling to 70B+ models used in production?

---

> ### Author Response · Authors · 2025-11-20
>
> We thank the reviewer for careful reading and thoughtful comments about our work. Below, we address the questions and concerns you have raised:
>
> ---
>
> **Q1: Statistical significance**
>
> **Can you provide confidence intervals or statistical tests for the main results in Tables 1-3? Given the small effect sizes, this is essential to determine if improvements are real or within noise margins.**
>
> Due to limited time during the rebuttal period, we reran all experiments with 5 independent seeds and now report \\(\text{mean} \pm \\text{std}\\) values.
>
> Updated CIFAR-10 and WMDP significance results appear in Tables 2 and 4 of the revised main paper.
>
> TOFU results (too large for the main tables) appear in Tables 13 and 14 in Section 7.7.1 of the appendix.
>
> These additional runs clarify the variance patterns and confirm that OFMU’s improvements remain consistent across seeds. We acknowledge that more extensive tests (e.g., paired t-tests over 10 runs) would further strengthen the conclusions and plan to include expanded significance analysis in the camera-ready version.
>
> ---
>
> **Q2: Computational overhead**
>
> **What is the actual training time and memory consumption of OFMU compared to baselines like GA and NPO? How many additional gradient computations does the Hessian-vector product require per iteration?**
>
> In the revised draft, we have added new Sections 7.4 and 7.5.13 (highlighted in red) for theoretical and empirical analyses clarifying the computational overhead and memory consumption of OFMU relative to baselines.
> Section 7.4.2 (lines 879--889) specifically explains why the Hessian-vector product is inexpensive: it is computed via Pearlmutter’s method, which requires only a single additional backward pass. As a result, the Hessian--vector product has approximately the same cost as a standard gradient evaluation.
> Empirically, OFMU incurs only a 2--4x overhead depending on the inner loop length \\(T\\), which matches our theoretical complexity of \\(\\Theta(K T B d)\\)
>
> ---
>
>
> **Q3: Scalability validation**
>
> **Have you tested on models larger than 7B parameters? If not, what are the expected computational bottlenecks that would prevent scaling to 70B+ models used in production?**
>
> We appreciate the reviewer’s interest in scalability. Our experiments were conducted on models up to 7B parameters due to the GPU memory required to load and run larger models. Scaling to 70B+ models is therefore constrained primarily by hardware availability (Section 5.1 line 351)  rather than by the OFMU method.
>
> ---
> We thank the reviewer again for their thoughtful comments. We hope our responses address the points raised and we welcome any further discussion or clarification.

---

### Official Review · Reviewer_R451 · 2025-11-01

**Soundness:** 3
**Presentation:** 2
**Contribution:** 3
**Rating:** 6
**Confidence:** 3

**Summary:**

`This paper proposes a bi-level optimization formulation for LLM unlearning, where the inner level optimizes the LLM paramater to unlearn forget data while  decorrelating forget data and retain data gradient and the outer level restores the unlearned model performance on retain data. To avoid heavy bi-level optimization, this paper proposes an iternating optimization scheme on mini-batch for stable training. This paper presents theoratical analysis and also shows performance improvement on LLM unlearning and image classifier unlearning.

**Strengths:**

* The bi-level optimization framework is reasonable and easy to follow.
* The involvement of decorrelating penalty term is well motivated and promising.

**Weaknesses:**

* Potential retain data robustness issue. The similarity-aware penalty relies on the cosine similarity between $\nabla_\theta L_f$ and $\nabla_\theta L_r$, which assumes meaningful gradient directions from the retain data. What's the effect of different retain data on the unlearning training?
* Concerning performance on WMDP. Table 3 in Appendix presents the performance for WMDP. From my understanding on WMDP dataset, Bio Acc. and Cyber Acc. are measuring how the unlearned LLM performs on the desired knowledge to remove (Bio/Cyber), but Table mentions higher the better, is this really correct?

**Questions:**

* Is the outer level optimization always fine-tuned to divergence? Algorithm 1 suggests only 1-step for outer loop.
* How is the penalty schedule determined in this work?
* Given that the framework operates in bi-level optimization, involving runtime information can better allow users to estimate its cost.

---

> ### Author Response · Authors · 2025-11-20
>
> We thank the reviewer for the detailed and constructive feedback of our work. Below we address each concern:
>
> ---
>
> **Weaknesses**:
>
> ---
> **Potential retain data robustness issue. The similarity-aware penalty relies on the cosine similarity between \\( \nabla_{\theta} L_{f} \\) and \\( \nabla_{\theta} L_{r} \\), which assumes meaningful gradient directions from the retain data. What is the effect of different retain data on the unlearning training?**
>
> The reviewer raises an important question regarding the effect of different retain data on the similarity aware penalty. In our formulation, the retain set \\(D_r\\) is fixed and corresponds to the portion of the original training distribution on which the model must preserve utility after unlearning. Exploring alternative definitions of “retain data” is not part of the standard unlearning setup, where \\(D_r\\) is predetermined by the task rather than a tunable choice.
> If the reviewer intended a different interpretation, we would appreciate clarification so we can address the question more precisely.
>
> ---
> **Concerning performance on WMDP. Table 3 in Appendix presents the performance for WMDP. From my understanding on WMDP dataset, Bio Acc. and Cyber Acc. are measuring how the unlearned LLM performs on the desired knowledge to remove (Bio/Cyber), but Table mentions higher the better, is this really correct?**
> We apologize for the confusion regarding the Bio and Cyber accuracy scores in Table 4 (revised draft). These metrics indeed measure the model’s ability to answer questions from hazardous domains that should be forgotten.
> To maintain consistency with MMLU and to enable clearer comparison, we report **unlearning efficacy** defined as
>
> unlearning\_efficacy = 1 - accuracy
>
>
>
> Higher values therefore indicate stronger removal of hazardous knowledge.
> We have updated the revised draft and explicitly clarified this in Section 7.5.1 (lines 917).
>
> ---
>
> **Questions:**
>
> ---
>
> **Is the outer level optimization always fine-tuned to divergence? Algorithm 1 suggests only 1-step for outer loop.**
>
> The outer optimization does not diverge; it converges to a stationary point of the penalty objective. While Algorithm 1 shows one outer step per iteration, this process is repeated for K iterations until convergence.  The mechanism is similar to adversarial training and meta-learning, where alternating single steps between two coupled objectives are sufficient (e.g., https://arxiv.org/pdf/1703.03400). Our convergence analysis (Lemma 3, Section 4, lines 328–340 in the revised version) and empirical results confirm that this two-loop algorithm—with multiple inner steps followed by a single outer step per cycle—is stable and effective.
>
> ---
>
> **How is the penalty schedule determined in this work?**
>
> The penalty coefficient follows a monotonically increasing schedule, as shown in Algorithm 1 and section 3.5 (line 261-265). We have added further explanation for the penalty schedule used in our experiments in Section 7.5.12 (lines 1242–1245).
>
> ---
>
> **Given that the framework operates in bi-level optimization, involving runtime information can better allow users to estimate its cost.**
>
> We appreciate this suggestion. The revised draft has detailed theoretical and empirical runtime analysis in Sections 7.4 and 7.5.13 (highlighted in red). These sections include comparisons against GA, NPO, and other baselines, along with complexity expressions and practical runtime measurements.
>
> ---
> We thank the reviewer again for their detailed feedback. We hope our responses address the reviewer's comments, and we welcome any further discussion or clarification.

---

> > ### Comment · Reviewer_R451 · 2025-11-24
> >
> > Thanks for the detailed resposne and clarification. All my concerns are addressed. I would like to keep my current rating. Please incorporate these incorporations in final version of the paper.

---

### Official Review · Reviewer_jsTy · 2025-11-01

**Soundness:** 3
**Presentation:** 3
**Contribution:** 3
**Rating:** 6
**Confidence:** 3

**Summary:**

This paper proposes OFMU, an LLM unlearning method based on bi-level optimization. Specifically, the authors formulate unlearning as a bi-level problem, where the lower level maximizes the forget loss and negative cosine similarity between the forget and retain gradients. The upper level minimizes the retain loss. For efficient optimization, the authors propose to convert the original problem into a single-level problem and leverage a two-loop optimization scheme to update the model in an efficient and stable way. The authors also provide theoretical analyses to show the convergence guarantee of the proposed method. Finally, experiments on both language and vision benchmarks demonstrate the effectiveness of the proposed method.

**Strengths:**

1. The paper formulates unlearning as a bi-level optimization problem, which addresses the issues in prior works where the forget and retain objectives are difficult to balance. The authors further propose a practical algorithm to solve the optimization problem.
2. Experiments on both language and vision tasks show that the proposed method outperforms the strong baselines.
3. The paper is well-written. The method is motivated and explained well.

**Weaknesses:**

1. The method introduces additional costs due to the two-loop optimization and the computation of the Hessian-vector product. How's the speed of the method compared to simple baselines such as GA or NPO?
2. The method introduces additional hyperparameters such as the penalty parameter and the regularization parameter $\beta$. How robust is the performance with respect to these hyperparameters?

**Questions:**

Please see weaknesses

---

> ### Author Response · Authors · 2025-11-20
>
> We sincerely thank the reviewer for their thoughtful and constructive feedback. We address each point below.
>
> ---
>
> **1. The method introduces additional costs due to the two-loop optimization and the computation of the Hessian-vector product. How's the speed of the method compared to simple baselines such as GA or NPO?**
>
> In the revised draft, we have added new Sections 7.4 and 7.5.13 (highlighted in red) providing theoretical and empirical analyses clarifying the computational overhead and memory consumption of OFMU relative to baselines.
> Section 7.4.2 (lines 879--889) specifically explains why the Hessian--vector product is inexpensive: it is computed via Pearlmutter’s method, which requires only a single additional backward pass. As a result, the Hessian--vector product has approximately the same cost as a standard gradient evaluation.
> Empirically, OFMU incurs only a 2--4x overhead depending on the inner loop length \\(T\\), which matches our theoretical complexity of \\(\\Theta(K T B d)\\).
>
> ---
>
> **2. The method introduces additional hyperparameters such as the penalty parameter and the regularization parameter \\(\\beta\\)
> . How robust is the performance with respect to these hyperparameters?**
>
> We have added hyperparameter ablations analyzing sensitivity to both the penalty coefficient \\(\\rho\\) and the regularization parameter \\(\\beta\\) in  Section 7.5.8 (Table 10) for TOFU  and Section 7.5.9 (Figure 5) for the CIFAR experiments. This demonstrates that the method is robust to these hyperparameter choices.
> Due to the time constraints of the rebuttal period, we were unable to include further ablations, although we plan to expand this analysis in the camera-ready version.
>
> ---
>
> We thank the reviewer once again for their thoughtful comments and constructive feedback. We hope that our responses sufficiently address all concerns, and we remain happy to provide any further clarification or engage in additional discussion.

---

### Official Review · Reviewer_vkLf · 2025-11-01

**Soundness:** 3
**Presentation:** 3
**Contribution:** 2
**Rating:** 4
**Confidence:** 4

**Summary:**

This work proposes a bi-level method to tackle the unlearning problem, treating forgetting as inner maximization and retention as outer minimization. A brief convergence theory is provided. Two models and 4 datasets are tested empirically.

**Strengths:**

This work tackles an important problem with a relatively new angle. The motivation is clear. The experiments are extensive. I think the 2-level approach makes sense, and the theoretical analysis supports it.

**Weaknesses:**

Some parts of this paper is not self-consistent and makes it hard to trust.

1. RMU (Risk Minimization Unlearning) is mis-cited.
RMU an important baseline throughout this work. However, it is attributed to Bu et al in Section 7.4.11 but also attributed to Li et al in Section 5.2. Both citations are wrong. Bu's method is termed NGDiff and has nothing to do with risk. Li's paper is actually about WMDP benchmark, not an optimization method. Also in Section 5.1, WMDP is again mis-cited to Bu et al.

In the end, what is RMU? Did you really implement NGDiff?

2. Tables are not trust-worthy.

I am highly suspicious of Table 1 to 4. First of all, the bold texts seem to suggest OFMU is the best, but this is simply misleading. In Table 5, **bold texts clearly mean the best performing method**. However, in Table 1 to 4, the authors always highlight OFMU even if it is not the best under many metrics. There are so many false emphasis so I will only point out a few: Table 1, LLaMA-3.2-1B-Instruct, FTR (RMU is better but OFMU is bolded); Table 2, Random Forgetting, RA and TA columns (FT and GA and IU are much better but OFMU is bolded).

Secondly and more importantly, Table 1 didn't not use fair baselines for comparison. The FQ column is almost all zeros and MU column is very low except OFMU, which is highly unlikely in the normal setting. I cannot list all entries in this big table that are suspicious to me, but I require clarity before I finalize my review score. What did I miss?

3. Learning rate concerns

There is only one learning rate given in Section 7.4.12. But your 2-level algorithm needs two learning rates, inner and outer. Why not give both learning rates? Did you discuss the extra cost of tuning learning rates? If one sweeps 5 values for one learning rate, one will need 5*5=25 values to sweep for two learning rates, which can be very inefficient and impractical.

**Questions:**

See weaknesses.

---

> ### Author Response · Authors · 2025-11-20
>
> We sincerely thank the reviewer for the careful reading, constructive feedback, and for pointing out issues that helped us improve the clarity and correctness of the paper. We address each concern below.
>
> ---
> **RMU (Risk Minimization Unlearning) is mis-cited. RMU an important baseline throughout this work. However, it is attributed to Bu et al in Section 7.4.11 but also attributed to Li et al in Section 5.2. Both citations are wrong. Bu's method is termed NGDiff and has nothing to do with risk. Li's paper is actually about WMDP benchmark, not an optimization method. Also in Section 5.1, WMDP is again mis-cited to Bu et al.
> In the end, what is RMU? Did you really implement NGDiff?**
>
> We sincerely apologize for the citation typos and the confusion they created. Thank you for catching them.
> RMU in our paper refers to the Representation Misdirection for Unlearning method introduced in the WMDP paper by Li et al, and this is the method we implemented and used throughout the experiments.
> In Section 5.2, the citation was correct, however, the appendix mistakenly attributed RMU to Bu et al., and we have corrected this now for both RMU and wmdp in the revised version.
> We initially used the term "risk minimization" because the WMDP paper and the RMU method primarily address the minimization of risks associated with hazardous knowledge. To avoid confusion, we have now updated to use the official name, "Representation Misdirection for Unlearning (RMU).
>
> ---
>
>
> **Tables are not trust-worthy.
> I am highly suspicious of Table 1 to 4. First of all, the bold texts seem to suggest OFMU is the best, but this is simply misleading. In Table 5, bold texts clearly mean the best performing method. However, in Table 1 to 4, the authors always highlight OFMU even if it is not the best under many metrics. There are so many false emphasis so I will only point out a few: Table 1, LLaMA-3.2-1B-Instruct, FTR (RMU is better but OFMU is bolded); Table 2, Random Forgetting, RA and TA columns (FT and GA and IU are much better but OFMU is bolded). Secondly and more importantly, Table 1 didn't not use fair baselines for comparison. The FQ column is almost all zeros and MU column is very low except OFMU, which is highly unlikely in the normal setting. I cannot list all entries in this big table that are suspicious to me, but I require clarity before I finalize my review score. What did I miss?**
>
> We understand the reviewer’s concern and have investigated the issue carefully.
>
> The very small scores for many baselines arise from the inherent imbalance between forgetting and retention objectives.
> Baselines like GA and NPO tend to collapse one objective to optimize the other, resulting in extremely poor performance on metrics that reflect the unaddressed objective. To ensure that these results were not an artifact or implementation bug, we re-ran all experiments with 5 independent seeds during the rebuttal period.
> The updated mean ± std values are provided in:
> Tables 2 and 4 in the main paper ( CIFAR-10 and WMDP ),
> Tables 13 and 14 in Section 7.7.1 (TOFU).
> The multi-seed results confirm the trends and validate that the low values are not noise but a consequence of how these baselines behave under conflicting objectives.
>
> To ensure full transparency, we will open-source all configuration files and code upon acceptance, making it possible to reproduce every result.
>
> ---
> **Learning rate concerns
> There is only one learning rate given in Section 7.4.12. But your 2-level algorithm needs two learning rates, inner and outer. Why not give both learning rates? Did you discuss the extra cost of tuning learning rates? If one sweeps 5 values for one learning rate, one will need 5*5=25 values to sweep for two learning rates, which can be very inefficient and impractical.**
>
>
> The reviewer is correct that a naive bi-level optimization could require tuning two separate learning rates, leading to a quadratic increase in the hyperparameter search cost. But OFMU uses a **single learning rate** for both loops to avoid costly hyperparameter tuning. This is feasible because:
> (i) The inner loop is short (typically \(T=5\) steps), acting as a preconditioning phase rather than a full optimization/convergence.
> (ii) The penalty reformulation \\(F(\theta)\\) creates a unified objective, making a single learning rate theoretically stable and empirically sufficient.
> So, by design of OFMU, single learning rate is sufficient.
>
> ---
>
> We sincerely thank the reviewer for their thoughtful and constructive feedback. We believe our responses resolve the concerns raised, and we are happy to provide any additional clarification if needed.

---

### Meta-Review · Area_Chair_DeBK · 2025-12-28

**Summary:**

This submission proposes OFMU, a penalty-based bi-level optimization framework for machine unlearning that enforces forgetting via an inner maximization objective while restoring utility through an outer minimization. Reviewers generally agree that the formulation is well motivated, theoretically grounded, and clearly presented, and the rebuttal satisfactorily addresses most reviewer-specific concerns, including citation issues, formatting, hyperparameter robustness, and additional runtime analysis.

In light of these strengths, I recommend acceptance.

Furthermore, I would like to highlight two points for clarification in the final version: (1) Computational cost. While the authors note that Hessian–vector products can be computed efficiently via Pearlmutter’s trick, the similarity-aware penalty appears to introduce additional backward passes on top of multiple inner-loop updates, which merits clearer accounting; and (2) Generality of the forgetting objective. The current instantiation assumes a loss-based forgetting signal (negative cross-entropy), and it would be helpful to clarify how the bi-level structure and gradient-similarity penalty extend to alternative forgetting objectives such as RMU or NPO/SimNPO.

**Reviewer Concerns:**

Concerns adequately addressed by the rebuttal: Citation and presentation issues, hyperparameter sensitivity, learning-rate tuning, and basic runtime clarification.

Concerns still outstanding: Clarifications on computational cost and forget objective function choice.

**Reviewer Scores:**

I think all scores will stay at 6.

---

### Decision · Program_Chairs · 2026-01-26

Accept (Poster)